# PruneFuse: Efficient Data Selection via Weight Pruning and Network Fusion

**Humaira Kousar**[*]                                                    *humairakousar32@kaist.ac.kr*
*School of Electrical Engineering*
*KAIST*

**Hasnain Irshad Bhatti**[*]                                             *hasnain@kaist.ac.kr*
*School of Electrical Engineering*
*KAIST*

**Jaekyun Moon**                                                         *jmoon@kaist.edu*
*School of Electrical Engineering and Kim Jaechul Graduate School of AI*
*KAIST*

**Reviewed on OpenReview:** *https://openreview.net/forum?id=BvnxenZwqY*

## Abstract

Efficient data selection is crucial for enhancing the training efficiency of deep neural networks and minimizing annotation requirements. Traditional methods often face high computational costs, limiting their scalability and practical use. We introduce PruneFuse, a novel strategy that leverages pruned networks for data selection and later fuses them with the original network to optimize training. PruneFuse operates in two stages: First, it applies structured pruning to create a smaller pruned network that, due to its structural coherence with the original network, is well-suited for the data selection task. This small network is then trained and selects the most informative samples from the dataset. Second, the trained pruned network is seamlessly fused with the original network. This integration leverages the insights gained during the training of the pruned network to facilitate the learning process of the fused network while leaving room for the network to discover more robust solutions. Extensive experimentation on various datasets demonstrates that PruneFuse significantly reduces computational costs for data selection, achieves better performance than baselines, and accelerates the overall training process.

## 1 Introduction

Deep learning models have achieved remarkable success across various domains, ranging from image recognition to natural language processing (Ren et al., 2015; Long et al., 2015; He et al., 2016). However, the performance of models heavily relies on access to large amounts of labeled data for training (Sun et al., 2017). In practical real-world applications, manually annotating massive datasets can be prohibitively expensive and time-consuming. Data selection techniques such as Active Learning (AL) (Gal et al., 2017) offer a promising solution to this challenge by iteratively selecting the most informative samples from the unlabeled dataset for annotation, thereby reducing labeling costs while approaching or even surpassing the performance of fully supervised training. Even with the rapid scaling of large language models and multimodal foundation models, effective adaptation to downstream tasks continues to demand high-quality, domain-aligned labeled data. A growing body of work demonstrates that principled selection techniques, including AL, outperform simple scaling of in-domain data in both final performance and overall computational efficiency (Xie et al., 2023; Yu et al., 2024; 2025). Traditional AL methods, however, incur severe computational overhead. Each selection

---

[*]Equal contribution.

cycle in AL typically requires extensive training or inference with a large model on the entire unlabeled pool. As model and dataset sizes grow, this repeated training becomes a critical scalability bottleneck, especially in resource-constrained environments. In this paper, we propose a novel strategy for efficient data selection in an AL setting that overcomes the limitations of traditional approaches. Our approach builds on the concept of model pruning, which selectively reduces the complexity of neural networks while preserving their accuracy. By utilizing small pruned networks as reusable data selectors, we eliminate the need to train large models, specifically during the data selection phase, thus significantly reducing computational demands. By enabling swift identification of the most informative samples, our method not only enhances the efficiency of AL but also ensures its scalability and cost-effectiveness in resource-limited settings. Additionally, we employ these pruned networks to train the final model through a fusion process, effectively harnessing the insights from the trained networks to accelerate convergence and improve generalization.

**Main Contribution.** To summarize, our key contribution is to introduce PruneFuse, an efficient and rapid data selection technique that leverages pruned networks. This approach mitigates the need for continuous training of a large model prior to data selection, which is inherent in conventional active learning methods. By employing pruned networks as data selectors, PruneFuse ensures computationally efficient selection of informative samples, which leads to overall superior generalization. Furthermore, we propose the novel concept of fusing these pruned networks with the original untrained model, enhancing model initialization and accelerating convergence during training.

We demonstrate the broad applicability of PruneFuse across various network architectures, providing researchers and practitioners with a flexible tool for efficient data selection in diverse deep learning settings. Extensive experimentation on CIFAR-10, CIFAR-100, Tiny-ImageNet-200, ImageNet-1K, text datasets (Amazon Review Polarity and Amazon Review Full), as well as Out-of-Distribution (OOD) benchmarks, shows that PruneFuse achieves superior performance to state-of-the-art AL methods while significantly reducing computational costs.

## 2 Related Work

**Data Selection.** Recent studies have explored techniques to improve the efficiency of data selection in deep learning. Approaches such as coreset selection (Sener & Savarese, 2018), BatchBALD (Kirsch et al., 2019), and Deep Bayesian Active Learning (Gal et al., 2017) aim to select informative samples using techniques like diversity maximization and Bayesian uncertainty estimation. Parallelly, the domain of active learning has unveiled strategies, such as uncertainty sampling (Shen et al., 2018; Sener & Savarese, 2018; Kirsch et al., 2019), expected model change-based approach (Freytag et al., 2014; Käding et al., 2016), snapshot ensembles Jung et al. (2023), and query-by-density (Sener & Savarese, 2018). These techniques prioritize samples that can maximize information gain, thereby enhancing model performance with minimal labeling effort. While these methods achieve efficient data selection, they still require training large models for the selection process, resulting in significant computational overhead. Other strategies, such as Gradient Matching (Killamsetty et al., 2021a) optimize this selection process by matching the gradients of a subset with the training or validation set based on the orthogonal matching algorithm, and (Killamsetty et al., 2021b) proposes meta-learning based approach for online data selection. SubSelNet (Jain et al., 2023) proposes to approximate a model that can be used to select the subset for various architectures without retraining the target model, hence reducing the overall overhead. However, it involves a pre-training routine, which is very costly and must be repeated for any change in data or model distribution.

Proxy-based selection methods such as SVP (Coleman et al., 2020) train a smaller proxy model (e.g., ResNet-20) as a data selector for a larger target model (e.g., ResNet-56). However, after selecting a subset, the proxy is discarded and the target is trained from scratch on the selected subset. Since the proxy and target architectures are typically different and not directly aligned, there is generally no canonical way to reuse trained proxy weights directly in the target, except indirectly, e.g., via distillation. PruneFuse differs in that the data selector model is obtained by structured pruning of the target model, yielding a channel-aligned subnetwork of the target. This structural alignment enables weight-aligned fusion, where the weights of the trained pruned model are directly copied into the corresponding coordinates of the dense original model, while with a generic proxy like SVP's, there is no such one-to-one mapping of the proxy to the target model.

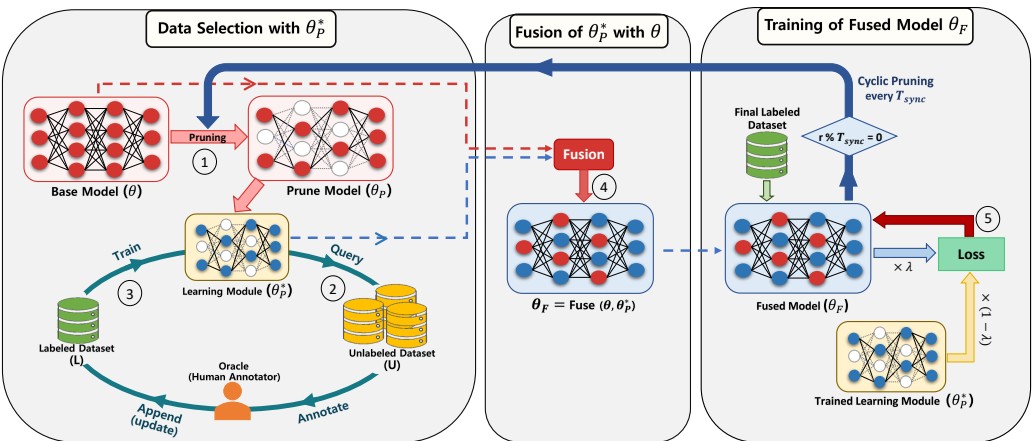

Figure 1: **Overview of the PruneFuse method**: (1) An untrained neural network is initially pruned to form a structured, pruned network $\theta_p$. (2) This pruned network $\theta_p$ queries the dataset to select prime candidates for annotation, similar to active learning techniques. (3) $\theta_p$ is then trained on these labeled samples to form the trained pruned network $\theta_p^*$. (4) The trained pruned network $\theta_p^*$ is fused with the base model $\theta$, resulting in a fused model. (5) The fused model is further trained on a selected subset of the data, incorporating knowledge distillation from $\theta_p^*$. At regular intervals $T_{\text{sync}}$, the fused model is utilized to dynamically update the pruned model for subsequent data selection.

In essence, fusion enables a warm-start for the target model in PruneFuse by leveraging the training compute of the selection process and results in faster convergence and better accuracy than training the dense target from scratch on the same selected subset.

**Efficient Deep Learning.** Efficient deep learning has gained significant attention in recent years. Methods such as Neural Architecture Search (NAS) (Zoph & Le, 2016; Wan et al., 2020), network pruning (Han et al., 2016), quantization (Dong et al., 2020; Jacob et al., 2018; Zhou et al., 2016), and knowledge distillation (Hinton et al., 2015; Yin et al., 2020) have been proposed to reduce model size and computational requirements. Neural network pruning has been extensively investigated as a technique to reduce the complexity of deep neural networks (Han et al., 2016). Pruning strategies can be broadly divided into Unstructured Pruning (Dong et al., 2017; Guo et al., 2016; Park et al., 2020) and Structured Pruning (Li et al., 2016; He et al., 2017; You et al., 2019; Ding et al., 2019) based on the granularity and regularity of the pruning scheme. Unstructured pruning often yields a superior accuracy-size trade-off, whereas structured pruning offers practical speedup and compression without necessitating specialized hardware. While pruning literature suggests pruning after training (Renda et al., 2020) or during training (Zhu & Gupta, 2017; Gale et al., 2019), recent research explores the viability of pruning at initialization (Lee et al., 2019; Tanaka et al., 2020; Frankle et al., 2021; Wang et al., 2020). In our work, we leverage the benefits of model pruning at initialization to create a small representative model for efficient data selection, allowing for the rapid identification of informative samples while minimizing computational requirements.

## 3 Background and Motivation

Efficient data selection is paramount in modern machine learning applications, especially when dealing with deep neural networks. We are given a labeled dataset $D = \{(x_i, y_i)\}_{i=1}^n$ drawn i.i.d. from an unknown distribution $p_Z$ over $\mathcal{X} \times \mathcal{Y}$, and a training procedure $\mathcal{A}$ that maps any labeled set $s \subseteq D$ to model parameters $\theta_s = \mathcal{A}(s)$. The subset selection problem can be framed as the challenge of selecting a subset $s$ of fixed size $b$ such that the model trained on $s$ has population risk close to that of the model trained on the full dataset:

$$s^\star \in \arg \min_{s \subseteq D, \, |s|=b} \left| E_{(x,y)\sim p_Z}\big[l(x,y;\theta_s)\big] - E_{(x,y)\sim p_Z}\big[l(x,y;\theta_D)\big] \right|, \tag{1}$$

where $\theta_s = \mathcal{A}(s)$ and $\theta_D = \mathcal{A}(D)$ denote the parameters obtained by training on the subset $s$ and the full dataset $D$, respectively.

### 3.1 Subset Selection Framework

Active Learning (AL) is a widely utilized iterative approach tailored for situations with abundant unlabeled data. Given a classification task with $C$ classes and a large pool of unlabeled samples $U$, AL revolves around selectively querying the most informative samples from $U$ for labeling. The process commences with an initial set of randomly sampled data $s^0$ from $U$, which is subsequently labeled. In subsequent rounds, AL augments the labeled set $L$ by adding newly identified informative samples. This cycle repeats until a predefined number of labeled samples, denoted by $b$, has been selected.

### 3.2 Network Pruning and Its Relevance

Network pruning emerges as a potent tool to reduce the complexity of neural networks. By eliminating redundant parameters, pruning preserves vital network functionalities while streamlining its architecture. Pruning strategies can be broadly categorized into Unstructured Pruning and Structured Pruning. Unstructured Pruning targets the removal of individual weights independent of their location. While it trims down the overall number of parameters, tangible computational gains on conventional hardware often demand very high pruning ratios (Park et al., 2017). On the other hand, Structured Pruning emphasizes the removal of larger constructs like kernels, channels, or layers. Its strength lies in preserving locally dense computations, which not only yields a leaner network but also bestows immediate performance improvements (Liu et al., 2017). Given its computational benefits, particularly in expediting evaluations and aligning with hardware optimizations, we opted for Structured Pruning over its counterpart. Importantly, pruned networks maintain the architectural coherence of the original model. This coherence makes them inherently more suitable for tasks such as data selection. Unlike heavily modified or entirely different models that can be used for data selection Coleman et al. (2020); Jain et al. (2023), the pruned model echoes the original structure, particularly advantageous in recognizing and prioritizing data samples that resonate with the patterns of the original network. The goal is clear: to develop a data selection strategy that conserves computational resources, minimizes memory overhead, and potentially improves model generalization.

## 4 PruneFuse

In this section, we delineate the PruneFuse methodology. The procedure begins with network pruning at initialization, offering a streamlined model for data selection. Upon attaining the desired data subset, the pruned model undergoes a fusion process with the original network, leveraging the structural coherence between them. The fused model is subsequently refined through knowledge distillation, enhancing its performance. An overall view of our proposed methodology is illustrated in Fig. 1.

Let $\theta_p \in \Theta_p$ denote a pruned model (obtained by structured pruning of the target architecture $\theta$), and let $\mathcal{S}_b(D; \theta_p)$ represents an acquisition operator that returns a subset of size $b$ by scoring/ranking examples in $D$ using $\theta_p$ (e.g., least-confidence, entropy, or greedy $k$-centers). PruneFuse selects $s_p = \mathcal{S}_b(D; \theta_p)$, $|s_p| = b$, and then trains the target model on $s_p$, i.e., $\theta_{s_p} = \mathcal{A}(s_p)$. This yields the following proxy-constrained variant of Eq. 1:

$$s_p^\star \in \arg\min_{s_p \subseteq D} \quad \left| \mathbb{E}_{(x,y)\sim p_Z}\big[l(x,y;\theta_{s_p})\big] - \mathbb{E}_{(x,y)\sim p_Z}\big[l(x,y;\theta_D)\big] \right|$$
$$\text{s.t.} \quad |s_p| = b, \qquad \exists \theta_p \in \Theta_p \text{ such that } s_p = \mathcal{S}_b(D; \theta_p). \tag{2}$$

where the subset can be defined as $s_p = \{(x_i, y_i) \in D : \text{score}(x_i; \theta_p) \geq \tau\}$ where $\tau$ is chosen so that $|s_p| = b$. Equivalently, for score-based acquisition (e.g., least-confidence or entropy), $s_p$ can be chosen by first computing $\text{score}(x; \theta_p)$ and then selecting the top-$b$ as $\mathcal{S}_b(D; \theta_p) = \text{Top}_b\{(x_i, y_i) \in D \text{ by } \text{score}(x_i; \theta_p)\}$. Whereas, for diversity-based acquisition (e.g., greedy $k$-centers), $\mathcal{S}_b$ denotes the corresponding greedy selection routine applied to embeddings/features produced by the pruned model.

Eq. 2 formalizes the goal of selecting $s_p$ so that training the target model on $s_p$ yields population risk close to training on the full dataset $D$, while performing selection using an efficient pruned model. Algorithm 1 precisely describes the PruneFuse methodology, i.e. training the proxy on the current labeled pool, scoring the unlabeled pool, and querying the next batch for annotation. The key insight is that structural coherence

between the pruned and target architectures makes this acquisition effective for the target while greatly reducing selection-time computation.

## 4.1 Pruning at Initialization

Pruning at initialization has been demonstrated to uncover superior solutions compared to the conventional approach of pruning an already trained network followed by fine-tuning (Wang et al., 2020). Specifically, it shows potential in training time reduction and enhanced model generalization. In our methodology, we employ structured pruning due to its benefits, such as maintaining the architectural coherence of the network, enabling more predictable resource savings, and often leading to better-compressed models in practice.

Consider an untrained neural network, represented as $\theta$. Let each layer $\ell$ of this network have feature maps or channels denoted by $c^\ell$, with $\ell \in \{1, \ldots, L\}$. Channel pruning results in binary masks $m^\ell \in \{0, 1\}^{d^\ell}$ for every layer, where $d^\ell$ represents the total number of channels in layer $\ell$. The pruned subnetwork, $\theta_p$, retains channels described by $c^\ell \odot m^\ell$, where $\odot$ symbolizes the element-wise product. The sparsity $p \in [0, 1]$ of the subnetwork illustrates the proportion of channels that are pruned: $p = 1 - \sum_\ell \mathbf{1}^\top m^\ell / \sum_\ell d^\ell$.

To reduce the model complexity, we employ the channel pruning procedure $prune(\theta, p)$, which prunes the model $\theta$ to a desired sparsity $p$. We first assign scores $z^\ell \in \mathbb{R}^{d^\ell}$ to every channel in the network according to their magnitude (using the L2 norm). The channels $C$ are represented as $(c^1, \ldots, c^L)$. Then we take the magnitude scores $Z = (z^1, \ldots, z^L)$ and translate them into masks $m^\ell$ such that the cumulative sparsity of the network, in terms of channels, is $p$. We employ a one-shot channel pruning that scores all the channels simultaneously based on their magnitude and prunes the network to $p$ sparsity. Although previous works suggest re-initializing the network to ensure proper variance (van Amersfoort et al., 2020), the performance gains are marginal; we therefore retain the weights of the pruned network before training.

## 4.2 Data Selection via Pruned Model

We begin by randomly selecting a small subset of data samples, denoted as $s^0$, from the unlabeled pool $U = \{x_i\}_{i \in [n]}$ where $[n] = \{1, \ldots, n\}$. These samples are then annotated. The pruned model $\theta_p$ is trained on this labeled subset $s^0$, resulting in the trained pruned model $\theta_p^*$. At each subsequent round, $\theta_p^*$ scores $U$ and proposes a batch of $k$ points for annotation.

We instantiate three widely used criteria for data selection, namely Least Confidence (LC) (Settles, 2012), Entropy (Shannon, 1948), and Greedy k-centers (Sener & Savarese, 2018).

1. LEAST CONFIDENCE based selection tends toward samples where the pruned model exhibits the least confidence in its predictions. The confidence score is essentially the highest probability the model assigns to any class label. Thus, the uncertainty score for a given sample $x_i$ based on LC is defined as $\text{score}(x_i; \theta_p^*)_{\text{LC}} = 1 - \max_{\hat{y}} P(\hat{y}|x_i; \theta_p^*)$. 2. In EN-TROPY based selection, the entropy of the model's predictions is the focal point. Samples with high entropy indicate situations where $\theta_p^*$ is ambivalent about the correct label. For each sample in $U$, the

---

**Algorithm 1** PruneFuse

**Notation**: Labeled dataset $L$, prune model $\theta_p$, fuse model $\theta_F$.

**Input**: Unlabeled dataset $U$, initial labeled data $s^0$, original model $\theta$, pruning ratio $p$, AL rounds $R$ to achieve budget $b$, synchronization interval $T_{sync}$, and acquisition score.

1: Randomly initialize $\theta$
2: $\theta_p \leftarrow \text{Prune}(\theta, p)$ //structured pruning
3: $\theta_p^* \leftarrow \text{Train}(\theta_p, s^0)$
4: $L \leftarrow s^0$
5: **for** $r = 1$ to $R$ **do**
6: $\quad D_k \leftarrow \text{Top}_k(\{\text{score}(\mathbf{x}; \theta_p^*) \mid \mathbf{x} \in U\})$
7: $\quad$ Query labels $y_k$ for selected samples $D_k$
8: $\quad L \leftarrow L \cup \{(D_k, y_k)\}$
9: $\quad$ **if** $T_{sync} = 0$ or $r \bmod T_{sync} \mathrel{!=} 0$ **then**
10: $\quad\quad \theta_p^* \leftarrow \text{Train}(\theta_p, L)$
11: $\quad$ **else if** $r \bmod T_{sync} = 0$ **then**
12: $\quad\quad \theta_F^* \leftarrow \text{Fuse}(\theta, \theta_p^*)$ and Fine-tune *(w/ KD)* on $L$
13: $\quad\quad \theta_p^* \leftarrow \text{Prune}(\theta_F^*, p)$ and Fine-tune on $L$

14: $\theta_F \leftarrow Fuse(\theta, \theta_p^*)$
15: $\theta_F^* \leftarrow \text{Fine-tune } \theta_F \text{ *(w/ KD)* on } L$

16: **Return** $L, \theta_F^*$

---

uncertainty based on entropy is computed as $\text{score}(x_i; \theta_p^*)_{\text{Entropy}} = -\sum_{\hat{y}} P(\hat{y}|x_i; \theta_p^*) \log P(\hat{y}|\mathbf{x}_i; \theta_p^*)$. Subsequently, we select the top-$k$ samples exhibiting the highest uncertainty scores, proposing them as prime candidates for annotation. 3. The objective of the GREEDY K-CENTERS algorithm is to cherry-pick $k$ centers

from the dataset such that the maximum distance of any sample from its nearest center is minimized. The algorithm proceeds in a greedy manner by selecting the first center arbitrarily and then iteratively selecting the next center as the point that is furthest from the current set of centers. The selection can be mathematically represented as $x = \arg\max_{x \in U} \min_{c \in \text{centers}} d(x, c)$ where centers is the current set of chosen centers and $d(x, c)$ is the distance between point $x$ and center $c$. Although various metrics can be used to compute this distance, we opt for the Euclidean distance since it is widely used in this context.

*Remark.* These criteria are standard, and our contributions are orthogonal to the choice of acquisition score. Alternative or learned scores can be seamlessly integrated into our pipeline; see Supplementary Materials 9.5, 9.13 for more details.

## 4.3 Training of Pruned Model

Once we have selected the samples from $U$, they are annotated to get their respective labels. These freshly labeled samples are assimilated into the labeled dataset $L$. At the start of each training cycle, a fresh pruned model $\theta_p$ is generated. Training from scratch in every iteration is vital to prevent the model from developing spurious correlations or overfitting to specific samples (Coleman et al., 2020). This further ensures that the model learns genuine patterns in the updated labeled dataset without carrying over potential biases from previous rounds. The training process adheres to a typical deep learning paradigm. Given the dataset $L$ with samples $(x_i, y_i)$, the aim is to minimize the loss function: $\mathcal{L}(\theta_p, L) = \frac{1}{|L|} \sum_{i=1}^{|L|} \mathcal{L}_i(\theta_p, x_i, y_i)$, where $\mathcal{L}_i$ denotes the individual loss for the sample $x_i$. Training unfolds over multiple iterations (or epochs). In each iteration, the weights of $\theta_p$ are updated using backpropagation with an optimization algorithm such as stochastic gradient descent (SGD).

This process is inherently iterative, as in standard Active Learning. After each round of training, new samples are chosen, annotated, and the model is reinitialized and retrained from scratch. This cycle persists until certain stopping criteria, e.g., labeling budget or desired performance, are met. With the incorporation of new labeled samples at every stage, $\theta_p^*$ progressively refines its performance, becoming better suited for the subsequent data selection phase.

## 4.4 Fusion with the Original Model

After achieving the predetermined budget, the next phase is to integrate the insights from the trained pruned model $\theta_p^*$ into the untrained original model $\theta$. This step is crucial, as it amalgamates the learned knowledge from the pruned model with the expansive architecture of the original model, aiming to harness the best of both worlds.

**Rationale for Fusion.** Traditional pruning and fine-tuning methods often involve training a large model, pruning it down, and then fine-tuning the smaller model. While this is effective, it does not fully exploit the potential benefits of the larger, untrained model. The primary reason is that the pruning process might discard useful structures and connections within the original model that were not yet leveraged during initial training. By fusing the trained pruned model with the untrained original model, we aim to create a model that combines the learned knowledge by $\theta_p^*$ with the broader, untrained model $\theta$.

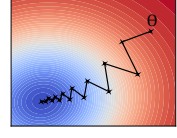

(a) $\theta$ trajectory

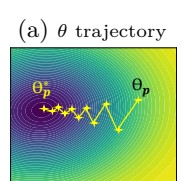

(b) $\theta_p$ trajectory

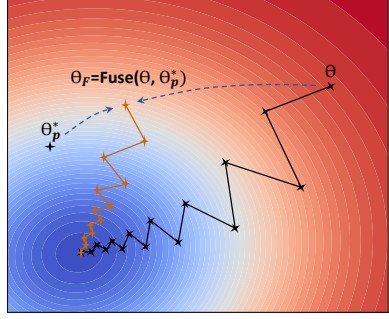

(c) $\theta_F$ with a refined trajectory due to fusion

Figure 2: **Evolution of training trajectories**. Conceptual illustration of how Pruning $\theta$ to $\theta_p$ tailors the loss landscape from 2a to 2b, allowing $\theta_p$ to converge on an effective configuration, denoted as $\theta_p^*$. This model, $\theta_p^*$, is later fused with the original $\theta$, which provides a better initialization and leads to an improved trajectory for $\theta_F$ to follow, as depicted in 2c.

**The Fusion Process.** Fusion transfers the trained parameters of the pruned selector $\theta_p^*$ into the *corresponding coordinates* of the original (untrained) dense model $\theta$, producing the fused model,

$$\theta_F = \text{Fuse}(\theta, \theta_p^*).$$

We view $\theta$ as a sequence of layers $j = 1, \ldots, L$. For each layer $j$, structured pruning at initialization selects a subset of output-channel indices $I_j \subseteq \{1, \ldots, C_{\text{out}}^{(j)}\}$ and, by architectural coherence, the pruned layer of $\theta_p^*$ is isomorphic to the sub-tensor of the dense layer indexed by $[I_j \times I_{j-1}]$ (with $I_0 = \{1, \ldots, C_{\text{in}}^{(1)}\}$, e.g., RGB channels). Instead of a coordinate copy, one may spread the trained pruned weights across multiple dense coordinates via a layer-wise dispersion map $D_j$:

$$W_F^{(j)} \leftarrow \underbrace{D_j\big(W_{p^*}^{(j)}\big)}_{\text{expanded onto } (C_{\text{out}}^{(j)}, C_{\text{in}}^{(j)})} \odot \mathbf{M}_j \; + \; W_{\text{init}}^{(j)} \odot (1 - \mathbf{M}_j),$$

where $\mathbf{M}_j$ is a binary mask that indicates where the dispersed weights land. Further details on the exact implementation of Fusion for Convolutional and Linear layers are provided in Supplementary Materials 9.4.

**Advantages of Retaining Unaltered Weights.** By copying the weights from the trained pruned model $\theta_p^*$ into their corresponding locations within the untrained original model $\theta$, and leaving the remaining weights of $\theta$ yet to be trained, we create a unique blend. The weights from $\theta_p^*$ encapsulate the knowledge acquired during training, providing a warm-start (better initialization). Meanwhile, the rest of the untrained weights in $\theta$ offer an element of randomness. We consider various initialization strategies for the remaining weights after fusion, including retaining initial weights, zero initialization and random re-initialization (as implemented in PruneFuse). We provide this ablation in Table 18 of the Supplementary Materials. Fig. 2 illustrates the conceptual transformation in training trajectories resulting from the fusion process. We empirically validate this in Fig. 4, where this fused initialization in the dense model $\theta_F$ results in better performance than training the original model in isolation.

It is important to highlight that fusion is fundamentally different from Knowledge Distillation (KD), where the information is transferred iteratively through an auxiliary training objective on outputs/logits from a teacher to a student model. In contrast, fusion in PruneFuse is a one-shot initialization technique that uniquely integrates the training compute spent on the selector into the target via aligned weight reuse. Based on the strategy discussed until now, we show that PruneFuse already outperforms baseline AL (results provided

Table 1: Components of PruneFuse.

| Component | Required |
|---|---|
| Pruning at Initialization | ✓ |
| Fusion (*weight transfer*) | ✓ |
| Synchronization interval (*T_sync*) | *Optional* |
| Knowledge Distillation | *Optional* |

in the Supplementary Materials 9.7). However, since our trained pruned model can act as a teacher for the target model, it is natural to integrate KD in PruneFuse. Hence, we use KD as an optional refinement in PruneFuse to further leverage the insights learned during the selection process.

## 4.5 Refinement of the Fused Model

During the fine-tuning of the fused model $\theta_F$, we use: i) Cross-Entropy Loss, which quantifies the divergence between the predictions of $\theta_F$ and the actual labels in dataset $L$, and ii) Distillation Loss (optional), which measures the difference in the softened logits of $\theta_F$ and $\theta_p^*$. These softened logits are derived by tempering logits of $\theta_p^*$, which in our case is the teacher model, with a temperature parameter before applying the softmax function. The composite loss for the fine-tuning phase is formulated as a weighted average of both losses. The iterative enhancement of $\theta_F$ is governed by: $\theta_F^{(t+1)} = \theta_F^{(t)} - \alpha \nabla_{\theta_F^{(t)}} (\lambda \mathcal{L}_{\text{Cross Entropy}}(\theta_F^{(t)}, L) + (1 - \lambda)\mathcal{L}_{\text{Distillation}}(\theta_F^{(t)}, \theta_p^*))$. Here, $\alpha$ represents the learning rate, while $\lambda$ functions as a coefficient to balance the contributions of the two losses. Incorporating KD in the fine-tuning phase aims to harness the insights of the pruned model $\theta_p^*$. By doing so, our objective is to ensure that the fused model $\theta_F$ not only retains the trained weights of the pruned model but also reinforces this knowledge iteratively, optimizing the performance of $\theta_F$ in subsequent tasks.

## 4.6 Iterative Pruning of Fused Model

PruneFuse introduces a strategy to dynamically update the pruned model, $\theta_p$, from the trained fused model $\theta_F^*$ at predefined intervals $T_{\text{sync}}$. In each AL cycle, the pruned model $\theta_p$, obtained by pruning a randomly

initialized network, is trained on the labeled dataset $L$ and subsequently employed to score the unlabeled data $U$. At every $T_{\text{sync}}$ cycle, the pruned model $\theta_p$ is obtained by pruning the trained fused model $\theta_F^*$, which is then fine-tuned with $L$ to get $\theta_p^*$ and later employed to score the $U$ in the subsequent rounds. By periodically synchronizing the pruned model with the fused model at regular $T_{\text{sync}}$ intervals, PruneFuse effectively balances computational efficiency with data selection precision. This iterative refinement process enables the pruned model to leverage the robust architecture of the fused model, allowing it to evolve dynamically with each cycle and leading to continuous performance improvements. As a result, PruneFuse achieves a better trade-off between accuracy and efficiency, enhancing the AL process while maintaining computational viability. Table. 1 summarizes the core components of the PruneFuse pipeline, distinguishing essential elements from optional design choices. Pruning at initialization and weight-aligned fusion form the core of PruneFuse, while synchronization frequency and knowledge distillation are optional components that trade off computation and performance in practice.

## 5 Error Decomposition for PruneFuse

We analyze a standard subset representativeness gap on the finite pool $D$ (akin to coreset-style analyses in Active Learning (Sener & Savarese, 2018)). Let $\theta^t$ denote the dense target model at cycle $t$ (i.e., the dense model after fusion/fine-tuning when synchronization is performed), let $\theta_p^t$ denote the pruned selector used for acquisition at cycle $t$, and let $s_p \subseteq D$ be the subset it selects, with $|D| = n$ and $|s_p| = m$. We study the discrepancy between the average loss of $\theta^t$ on $s_p$ and on the full dataset $D$:

$$\left| \mathbb{E}_{(x,y) \in s_p} l(x, y; \theta^t) - \mathbb{E}_{(x,y) \in D} l(x, y; \theta^t) \right| = \left| \frac{1}{m} \sum_{(x_i, y_i) \in s_p} l(x_i, y_i; \theta^t) - \frac{1}{n} \sum_{i=1}^{n} l(x_i, y_i; \theta^t) \right|. \tag{3}$$

**Theorem 5.1.** *Under the Assumptions stated in Sec. 9.2 of Supplementary Materials, with probability at least* $1 - \eta$,

$$\left| \mathbb{E}_{(x,y) \in s_p} l(x, y; \theta^t) - \mathbb{E}_{(x,y) \in D} l(x, y; \theta^t) \right| \leq \delta + 2L\rho_t. \tag{4}$$

The bound decomposes the representativeness error into two terms: an intrinsic acquisition error ($\delta$) and an additional proxy–target mismatch term ($2L\rho_t$). PruneFuse uses synchronization to promote proxy–target alignment and help control this mismatch, so that the selection by the pruned model more closely resembles selection performed with the target model. Assumptions and proof of the bound, along with further discussion and empirical validation, are provided in the Supplementary Materials (Sec. 9.2 and Table 8).

## 6 Experiments

### 6.1 Experimental Setup

**Datasets.** The effectiveness of our approach is assessed on different image classification datasets: CIFAR-10 (Krizhevsky et al., 2009), CIFAR-100 (Krizhevsky et al., 2009), TinyImageNet-200 (Le & Yang, 2015), and ImageNet-1K (Deng et al., 2009). CIFAR-10 is partitioned into 50,000 training and 10,000 test samples, CIFAR-100 contains 100 classes and has 500 training and 100 testing samples per class, whereas TinyImageNet-200 contains 200 classes with 500 training, 50 validation, and 50 test samples per class. ImageNet-1K consists of 1,000 classes with approximately 1.2 million training images and 50,000 validation images, providing a comprehensive benchmark for evaluating large-scale image classification models. We also extend our experiments to text datasets (Amazon Review Polarity and Amazon Review Full) (Zhang & LeCun, 2015; Zhang et al., 2015) and to out-of-distribution (OOD) benchmark to assess generalization (results provided in Supplementary Materials 9.5).

**Implementation Details.** We used various model architectures: ResNet (ResNet-50, ResNet-56, ResNet-110, and ResNet-164), Wide-ResNet, VDCNN, and Vision Transformers (ViT) in our experiments. We pruned these architectures using the Torch-Pruning library (Fang et al., 2023) for different pruning ratios $p = 0.5, 0.6, 0.7$, and $0.8$ to get the pruned architectures. We ran these experiments for 181 epochs following the

Table 2: **Performance comparison** of Baseline and PruneFuse on CIFAR-10, CIFAR-100, Tiny ImageNet-200 and ImageNet-1K. This table summarizes the *top-1 test accuracy* of the final model (original in case of *AL* and Fused in *PruneFuse*) and *computational cost* of the data selector (in terms of FLOPs) for various pruning ratios ($p$) and labeling budgets($b$). *Params* corresponds to the number of parameters of the data selector model. All results use Least-Confidence sampling with $T_{\text{sync}} = 0$. ResNet-56 is utilized for CIFAR-10/100, while ResNet-50 is used for Tiny-ImageNet and ImageNet-1K. Results better than the Baseline are highlighted in **Bold**.

| | | ↑ Accuracy (%) | | | | | ↓ Computation ($\times 10^{16}$) | | | | |
|---|---|---|---|---|---|---|---|---|---|---|---|
| *Budget (b)* | | *10%* | *20%* | *30%* | *40%* | *50%* | *10%* | *20%* | *30%* | *40%* | *50%* |
| Method | Params | CIFAR-10 | | | | | | | | | |
| Baseline (AL) | 0.85 M | 80.53±0.20 | 87.74±0.15 | 90.85±0.11 | 92.24±0.16 | 93.00±0.11 | 0.31 | 1.76 | 4.64 | 8.94 | 14.66 |
| PruneFuse $p$=0.5 | 0.21 M | **80.92±0.41** | **88.35±0.33** | **91.44±0.15** | **92.77±0.03** | **93.65±0.14** | 0.08 | 0.44 | 1.16 | 2.24 | 3.67 |
| PruneFuse $p$=0.6 | 0.14 M | **80.58±0.33** | **87.79±0.20** | **90.94±0.13** | **92.58±0.31** | **93.08±0.42** | 0.05 | 0.28 | 0.74 | 1.43 | 2.35 |
| PruneFuse $p$=0.7 | 0.08 M | 80.19±0.45 | **87.88±0.05** | 90.70±0.21 | **92.44±0.24** | **93.40±0.11** | 0.03 | 0.16 | 0.42 | 0.80 | 1.32 |
| PruneFuse $p$=0.8 | 0.03 M | 80.11±0.28 | 87.58±0.14 | 90.50±0.08 | **92.42±0.41** | **93.32±0.14** | 0.01 | 0.07 | 0.18 | 0.36 | 0.58 |
| Method | Params | CIFAR-100 | | | | | | | | | |
| Baseline (AL) | 0.86 M | 35.99±0.80 | 52.99±0.56 | 59.29±0.46 | 63.68±0.53 | 66.72±0.33 | 0.31 | 1.77 | 4.67 | 9.00 | 14.76 |
| PruneFuse $p$=0.5 | 0.22 M | **40.26±0.95** | **53.90±1.06** | **60.80±0.44** | **64.98±0.40** | **67.87±0.17** | 0.08 | 0.44 | 1.17 | 2.26 | 3.70 |
| PruneFuse $p$=0.6 | 0.14 M | **37.82±0.83** | 52.65±0.40 | **60.08±0.22** | **63.70±0.25** | **66.89±0.46** | 0.05 | 0.28 | 0.75 | 1.44 | 2.36 |
| PruneFuse $p$=0.7 | 0.08 M | **36.76±0.63** | 52.15±0.53 | **59.33±0.17** | 63.65±0.36 | **66.84±0.43** | 0.03 | 0.16 | 0.42 | 0.81 | 1.34 |
| PruneFuse $p$=0.8 | 0.04 M | **36.49±0.20** | 50.98±0.54 | 58.53±0.50 | 62.87±0.13 | 65.85±0.32 | 0.01 | 0.07 | 0.19 | 0.37 | 0.60 |
| Method | Params | Tiny-ImageNet-200 | | | | | | | | | |
| Baseline (AL) | 23.9 M | 14.86±0.11 | 33.62±0.52 | 43.96±0.22 | 49.86±0.56 | 54.65±0.38 | 0.50 | 2.73 | 7.11 | 13.64 | 22.32 |
| PruneFuse $p$=0.5 | 6.10 M | **18.71±0.21** | **39.70±0.31** | **47.41±0.20** | **51.84±0.10** | **55.89±1.21** | 0.13 | 0.70 | 1.81 | 3.48 | 5.69 |
| PruneFuse $p$=0.6 | 3.92 M | **19.25±0.72** | **38.84±0.70** | **47.02±0.30** | **52.09±0.29** | **55.29±0.28** | 0.08 | 0.45 | 1.16 | 2.23 | 3.66 |
| PruneFuse $p$=0.7 | 2.24 M | **18.32±0.95** | **39.24±0.75** | **46.45±0.58** | **52.02±0.65** | **55.63±0.55** | 0.05 | 0.26 | 0.67 | 1.28 | 2.09 |
| PruneFuse $p$=0.8 | 1.02 M | **18.34±0.93** | **37.86±0.42** | **47.15±0.31** | **51.77±0.40** | **55.18±0.50** | 0.02 | 0.12 | 0.30 | 0.58 | 0.95 |
| Method | Params | ImageNet-1K | | | | | | | | | |
| Baseline (AL) | 25.5 M | 52.97±0.20 | 64.52±0.46 | 69.30±0.15 | 71.98±0.11 | 73.56±0.16 | 6.88 | 37.34 | 97.28 | 186.70 | 305.60 |
| PruneFuse $p$=0.5 | 6.91 M | **55.03±0.33** | **65.12±0.31** | **69.72±0.17** | **72.07±0.28** | **73.86±0.55** | 1.86 | 10.10 | 26.30 | 50.47 | 82.62 |
| PruneFuse $p$=0.6 | 4.59 M | **54.69±0.93** | **65.13±0.55** | **69.74±0.38** | **72.48±0.33** | **74.00±0.68** | 1.24 | 6.71 | 17.47 | 33.53 | 54.88 |
| PruneFuse $p$=0.7 | 2.74 M | **53.73±0.71** | 64.43±0.65 | 68.95±0.41 | 71.81±0.31 | **73.84±0.29** | 0.74 | 4.00 | 10.43 | 20.01 | 32.76 |
| PruneFuse $p$=0.8 | 1.35 M | **53.08±0.22** | 64.00±0.17 | 69.00±0.90 | 71.79±0.81 | **73.64±0.52** | 0.36 | 1.97 | 5.14 | 9.86 | 16.14 |

setup in Coleman et al. (2020) for CIFAR-10 and CIFAR-100 and for 100 epochs for TinyImageNet-200 and ImageNet-1K. We used a mini-batch of 128 for CIFAR-10 and CIFAR-100 and 256 for TinyImageNet-200 and ImageNet-1K. Further details are provided in Supplementary Materials 9.3. Initially, we start by randomly selecting 2% of the data. For the first round, we add 8% from the unlabeled set, then 10% in each subsequent round, until the required budget $b$ is met. After each round, we retrain the models from scratch, as described in the methodology. All experiments were carried out independently three times, and the mean is reported. Detailed experiments on various model architectures, datasets, labeling budgets, and data selection metrics are provided in Supplementary Materials 9.5. We also provide detailed Complexity Analysis and Error Decomposition for PruneFuse in Supplementary Materials 9.1 and 9.2, respectively.

## 6.2   Results and Discussions

**Main Experiments.** Table 2 benchmarks PruneFuse against the standard AL pipeline across different datasets. PruneFuse attains comparable or higher top-1 accuracy while consuming only a fraction of the computational resources measured in terms of FLOPs when computed for the whole training duration of the pruned network and the selection process for different label budgets. On CIFAR-10, with a pruned model ($p$=0.7), PruneFuse achieves similar or superior performance compared to the baseline, while reducing selector cost by more than 90% (e.g. $1.32 \times 10^{16}$ vs. $14.66 \times 10^{16}$ FLOPs at $b$=50%). Similarly, on CIFAR-100 with ($p = 0.5$ at $b$=50%), PruneFuse outperforms baseline's accuracy by 1% while reducing 73% of the computational costs. The advantage becomes even more pronounced on the larger benchmarks. For Tiny-ImageNet, an aggressively pruned selector ($p$=0.8) lifts accuracy from 54.65% to 55.18% while reducing selector computation by 96%. Similarly, on ImageNet-1K, with the same pruning ratio, PruneFuse attains

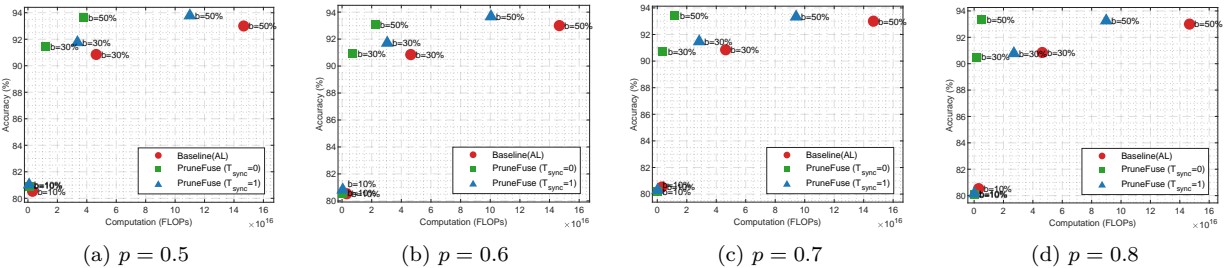

| (a) $p = 0.5$ | (b) $p = 0.6$ | (c) $p = 0.7$ | (d) $p = 0.8$ |

Figure 3: **Accuracy-cost trade-off** for PruneFuse. This figure illustrates the total number of FLOPs utilized by PruneFuse for data selection, compared to the baseline Active Learning method, for $T_{\text{sync}}=0, 1$ with labeling budgets $b = 10\%, 30\%, 50\%$. The experiments are conducted on the CIFAR-10 dataset using the ResNet-56 architecture. Subfigures (a), (b), (c), and (d) correspond to different pruning ratios of 0.5, 0.6, 0.7, and 0.8, respectively.

Table 3: **Comparison with baselines:** Final *top-1 test accuracy* and cumulative selector *cost* for different label budgets compared with baselines for ResNet-56 on CIFAR-10.

| Method | Params | ↑ Accuracy (%) | | | | | ↓ Computation ($\times 10^{16}$ FLOPs) | | | | |
|---|---|---|---|---|---|---|---|---|---|---|---|
| | *Budget (b)* | *10%* | *20%* | *30%* | *40%* | *50%* | *10%* | *20%* | *30%* | *40%* | *50%* |
| Baseline (AL) | 0.85 M | 80.53±0.20 | 87.74±0.15 | 90.85±0.11 | 92.24±0.16 | 93.00±0.11 | 0.31 | 1.76 | 4.64 | 8.94 | 14.66 |
| SVP | 0.26 M | 80.76±0.70 | 87.31±0.56 | 90.77±0.45 | 92.59±0.25 | 92.95±0.33 | 0.10 | 0.56 | 1.47 | 2.84 | 4.66 |
| PruneFuse ($T_{\text{sync}}=0$) | 0.21 M | **80.92±0.41** | **88.35±0.33** | **91.44±0.15** | **92.77±0.03** | **93.65±0.14** | **0.08** | **0.44** | **1.16** | **2.24** | **3.67** |
| PruneFuse ($T_{\text{sync}}=2$) | 0.21 M | **80.90±0.21** | **88.40±0.46** | **91.55±0.63** | **93.05±0.36** | **93.74±0.44** | **0.08** | 1.17 | 1.89 | 5.14 | 6.58 |
| PruneFuse ($T_{\text{sync}}=1$) | 0.21 M | **81.02±0.46** | **88.52±0.37** | **91.76±0.08** | **93.15±0.18** | **93.78±0.45** | **0.08** | 1.17 | 3.34 | 6.60 | 10.94 |
| BALD | 0.85 M | 80.61±0.24 | 88.11±0.41 | 91.21±0.56 | 92.98±0.81 | 93.36±0.62 | 0.34 | 1.81 | 4.71 | 9.03 | 14.77 |
| PruneFuse ($T_{\text{sync}}=1$) + BALD | 0.21 M | **80.71±0.46** | **88.38±0.37** | **91.44±0.55** | **93.16±0.21** | **93.58±0.07** | **0.08** | **1.18** | **3.36** | **6.62** | **10.97** |
| ALSE | 0.85 M | 80.73±0.32 | 88.13±0.41 | 90.99±0.56 | 92.58±0.63 | 93.13±0.75 | 0.41 | 1.96 | 4.92 | 9.29 | 15.08 |
| PruneFuse ($T_{\text{sync}}=0$) + ALSE | 0.21 M | **80.80±0.51** | **88.17±0.35** | **91.43±0.45** | **93.02±0.55** | **93.19±0.61** | **0.10** | **0.49** | **1.23** | **2.32** | **3.70** |

73.64% (baseline 73.56%) yet requires 95% less computation. These results show that PruneFuse achieves a superior accuracy–cost trade-off compared to a typical AL pipeline.

We further investigated how the synchronization interval $T_{sync}$ shapes the accuracy–cost trade-off. Fig. 3 plots top-1 accuracy versus cumulative selector FLOPs for four pruning ratios using $T_{sync} = 1$ (the pruned selector is updated from the finetuned fused model after every AL round). In all cases, PruneFuse lies on a superior accuracy–cost curve compared with the AL baseline. Even the lightest variant ($p$=0.8) achieves baseline-level accuracy while spending only one-tenth of the computation, and the configuration with $T_{sync} = 1$ delivers the best overall trade-off.

While PruneFuse introduces additional engineering components, such as structured pruning (at initialization or synchronization) and fusion during training, their associated costs are explicitly bounded and amortized over multiple active learning rounds (Section 9.1). In practice, this overhead is outweighed by substantial reductions in cumulative selector cost together with improved accuracy, as evidenced by the results in Table 2, Table 3 and Fig. 3.

**Comparison with Baselines.** Table 3 compares PruneFuse with several prominent active learning baselines, including SVP, ALSE (Jung et al., 2023), and BALD. We evaluate all methods under a *canonical* protocol: they share the same target architecture (e.g. ResNet-56) for final training and evaluation, the same labeled budgets, and the same target training schedule. Baselines follow the acquisition procedures described in the original works and are adapted only to fit this shared pipeline. The methods therefore differ only in the selector used for acquisition, as defined by each approach: SVP uses a smaller proxy network, whereas BALD and ALSE use the full target model as the selector. Specifically, SVP employs ResNet-20 (0.26M parameters) as the closest (parameter-wise) standard proxy to PruneFuse, which uses a 50% pruned ResNet-56 (0.21M parameters) as its selector. ALSE utilizes 5 snapshots of the data selector model at various training steps for data selection, and BALD uses Bayesian uncertainty with the full target model for selection. Results demonstrate that PruneFuse consistently outperforms SVP across all label budgets. For example, PruneFuse

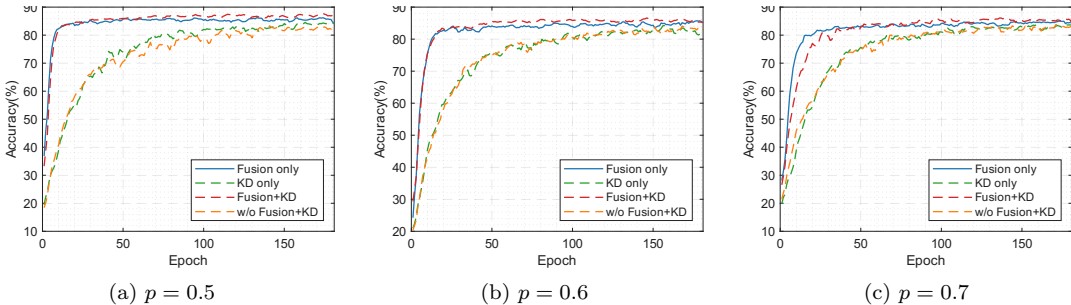

Figure 4: **Impact of Model Fusion on PruneFuse performance:** This figure compares the accuracy over epochs for different training variants within the PruneFuse framework on CIFAR-10 with ResNet-56. We compare fusion only, knowledge distillation (KD) only, fusion with KD, and training without fusion and KD. Subfigures (a), (b), and (c) correspond to $p = 0.5$, $0.6$, and $0.7$, respectively, for $b = 30\%$.

Table 4: **Evaluation on Coreset Selection task:** Baseline vs. PruneFuse ($p = 0.5$ and $T_{sync} = 0$) with Various Selection Metrics including Forgetting Events (Toneva et al., 2019), Moderate (Xia et al., 2023), and CSS (Zheng et al., 2023) on CIFAR-10 dataset using ResNet-56 architecture.

| Method | Model Params | | | Selection Metric | | | | |
| | Data Selector | Target Model | Budget ($b$) | Entropy | Least Conf. | Forgetting Events | Moderate | CSS |
|---|---|---|---|---|---|---|---|---|
| **Baseline** | 0.85M | 0.85M | 25% | 86.13±0.41 | 86.50±0.21 | 86.01±0.71 | 86.27±0.65 | 87.21±0.68 |
| **PruneFuse** | 0.21M | 0.85M | | **86.71±0.44** | **86.68±0.42** | **87.84±0.09** | **87.63±0.31** | **88.85±0.29** |
| **Baseline** | 0.85M | 0.85M | 50% | 91.41±0.21 | 91.28±0.35 | 93.31±0.76 | 90.97±0.55 | 90.68±0.35 |
| **PruneFuse** | 0.21M | 0.85M | | **92.24±0.45** | **92.75±0.65** | **93.40±0.19** | **91.08±0.49** | **90.79±0.51** |

peaks at 93.65% at $b$=50% compared to SVP with 92.95%, while having significantly lower computational costs (21% lower than SVP and 75% lower than the baseline). PruneFuse with iterative pruning of the fused model shows even better performance, reaching 93.78% and 93.74% accuracy with $T_{sync} = 1$ and 2 at 50% label budget, respectively, offering a trade-off between computational efficiency and accuracy. BALD demonstrates competitive results at higher label budgets (e.g., 93.36% at $b$=50%). However, BALD can be seamlessly integrated with PruneFuse. Capitalizing on the strengths of both methods, PruneFuse + BALD yields improved performance of 93.58% at 50% label budget while consuming 26% less computation. Similarly, PruneFuse + ALSE also results in better performance while having 4× less computation compared to ALSE.

Table 4 compares the performance of PruneFuse on the Coreset Selection task against various recent works. In this setup, the network is first trained on the entire dataset and then identifies a representative subset of data (coreset) based on the selection metric. The accuracy of the target model trained on that selected coreset is reported. The results show that PruneFuse seamlessly integrates with these advanced selection metrics, achieving competitive or superior performance compared to the baselines while being computationally inexpensive. This highlights the versatility of PruneFuse in enhancing existing coreset selection techniques.

**Additional Experiments and Ablation Studies.** Table 5 demonstrates results for Vision Transformers (21M params). Results show that PruneFuse consistently outperforms AL baseline across all label budgets for both CIFAR-10 and CIFAR-100, despite using small selector models. On CIFAR-10, PruneFuse with $p$=0.5 yields strong gains at low budgets (e.g., +5.69 points at $b$=10%) and maintains improvements even at higher budgets (e.g., +3.05 points at $b$=50%).

Table 5: **Performance Comparison** of Baseline and PruneFuse on CIFAR-10 and CIFAR-100 with **Vision Transformers (ViT)**. This table summarizes the test accuracy of final models (original in case of AL and Fused in PruneFuse) for various pruning ratios ($p$) and labeling budgets ($b$).

| Method | CIFAR-10 | | | | | CIFAR-100 | | | | |
| | Label Budget ($b$) | | | | | Label Budget ($b$) | | | | |
| | 10% | 20% | 30% | 40% | 50% | 10% | 20% | 30% | 40% | 50% |
|---|---|---|---|---|---|---|---|---|---|---|
| Baseline ($AL$) | 53.63 | 65.62 | 71.09 | 76.86 | 80.59 | 36.59 | 52.19 | 58.59 | 63.8 | 64.02 |
| PruneFuse ($p = 0.5$) | **59.32** | **71.63** | **75.61** | **81.50** | **83.64** | **46.84** | **57.16** | **62.56** | **65.8** | **67.75** |
| PruneFuse ($p = 0.6$) | 57.80 | 70.45 | 75.22 | 80.22 | 82.77 | 45.60 | 56.04 | 61.09 | 65.01 | 67.24 |
| PruneFuse ($p = 0.7$) | 56.17 | 69.25 | 73.87 | 79.47 | 82.15 | 44.46 | 55.36 | 60.99 | 64.47 | 66.85 |

The gains are more visible on CIFAR-100 (+10.25 points at $b$=10%) and offers consistent 3–4 point improvements at larger budgets.

We also extend our evaluation beyond vision tasks. Table 6 delineates experiments on text classification using VDCNN for Amazon Review Polarity and Amazon Review Full. PruneFuse again improves on the AL baseline in all pruning ratios and label budgets. In the case of Amazon Review Polarity, PruneFuse delivers consistent gains (e.g., $94.13 \rightarrow 94.66\%$ at $b$=10%, and $95.71 \rightarrow 95.87\%$ at $b$=50%). On the other hand, the improvements are greater for the Amazon Full dataset: with $p$=0.5, PruneFuse improves the baseline by 0.8–1.0 points across all budgets, and even with $p$=0.8 it continues to match or surpass the unpruned model.

Table 6: **Performance comparison** of Baseline and PruneFuse on Amazon Review Polarity Dataset and Amazon Review Full Dataset with VDCNN Architecture. This table summarizes the test accuracy of final models (original in case of AL and Fused in PruneFuse) for various pruning ratios ($p$) and labeling budgets ($b$).

| Method | Amazon Review Polarity | | | | | Amazon Review Full | | | | |
|---|---|---|---|---|---|---|---|---|---|---|
| | Label Budget ($b$) | | | | | Label Budget ($b$) | | | | |
| | 10% | 20% | 30% | 40% | 50% | 10% | 20% | 30% | 40% | 50% |
| Baseline (*AL*) | 94.13 | 95.06 | 95.54 | 95.73 | 95.71 | 58.46 | 60.65 | 61.50 | 62.20 | 62.43 |
| PruneFuse ($p = 0.5$) | **94.66** | **95.45** | **95.71** | **95.87** | **95.87** | **59.45** | **61.28** | **62.02** | **62.66** | **62.84** |
| PruneFuse ($p = 0.6$) | **94.62** | **95.38** | **95.69** | **95.82** | **95.88** | **59.28** | **61.14** | **62.08** | **62.62** | **62.81** |
| PruneFuse ($p = 0.7$) | **94.47** | **95.43** | **95.71** | **95.83** | **95.84** | **59.42** | **61.05** | **61.98** | **62.48** | **62.85** |
| PruneFuse ($p = 0.8$) | **94.33** | **95.37** | **95.63** | **95.79** | **95.85** | **59.24** | **61.05** | **61.94** | **62.45** | **62.77** |

Fig. 4 provides a joint component ablation that disentangles the effects of weight-aligned fusion and knowledge distillation (KD). Specifically, for each pruning ratio, we train the same dense target model under an identical training schedule on the data selected by the pruned model, and compare four variants: (i) No fusion / No KD, (ii) KD only, (iii) Fusion w/o KD, and (iv) Fusion + KD. Across pruning ratios, fusion (with or without KD) consistently accelerates convergence relative to training from scratch on the same selected subset, indicating that the primary gain arises from reusing the trained selector parameters as a warm-start initialization. KD provides an additional, complementary improvement in several settings, but is not required for PruneFuse to outperform training-from-scratch on the same selected subset. Further implementation details and additional ablations are provided in the Supplementary Materials (Sec. 9.6).

Table 7 demonstrates the impact of different pruning techniques (e.g., static pruning, dynamic pruning) and pruning criteria (e.g., L2 norm, GroupNorm Importance, LAMP Importance Fang et al. (2023)) on the performance of PruneFuse. Static pruning involves pruning the entire network at once at the start of training, whereas dynamic pruning incrementally prunes the network in multiple steps during training. In our implementation of dynamic pruning, the network is pruned in five steps over the course of 20 epochs. Results indicate that PruneFuse exhibits only minor performance variations (generally within 1–2% across label budgets), demonstrating that PruneFuse is highly flexible to various pruning strategies and criteria while maintaining strong performance in data selection tasks.

Table 7: **Effect of Pruning techniques and Pruning criteria** on PruneFuse ($p = 0.5$) on CIFAR-10 with ResNet-56.

| Method | Pruning Criteria | Label Budget ($b$) | | | | |
|---|---|---|---|---|---|---|
| | | 10% | 20% | 30% | 40% | 50% |
| **Baseline (AL)** | - | 80.53 | 87.74 | 90.85 | 92.24 | 93.00 |
| **PruneFuse** ($T_{sync} = 0$) (Dynamic Pruning) | Magnitude Imp. | 79.73 | 87.16 | **91.08** | **92.29** | 93.19 |
| | GroupNorm Imp. | 80.10 | **88.25** | 91.01 | 92.25 | 93.74 |
| | LAMP Imp. | **81.51** | 87.45 | 90.64 | **92.41** | 93.25 |
| **PruneFuse** ($T_{sync} = 0$) (Static Pruning) | Magnitude Imp. | **80.92** | 88.35 | 91.44 | 92.77 | 93.65 |
| | GroupNorm Imp. | 80.84 | 88.20 | 91.19 | 93.01 | 93.03 |
| | LAMP Imp. | 81.10 | 88.37 | 91.32 | 93.02 | 93.08 |
| **PruneFuse** ($T_{sync} = 1$) (Static Pruning) | Magnitude Imp. | 81.23 | 88.52 | 91.76 | 93.15 | 93.78 |
| | GroupNorm Imp. | 81.09 | 88.77 | 91.77 | 93.19 | 93.68 |
| | LAMP Imp. | 81.86 | 88.51 | 92.10 | 93.02 | 93.63 |

# 7 Conclusion

In this work, we present PruneFuse, a novel strategy that integrates pruning with network fusion to optimize the data selection pipeline for deep learning. PruneFuse leverages a small pruned model for data selection, which then seamlessly fuses with the original model, providing faster training, better generalization, and significantly reduced computational costs. It consistently outperforms existing baselines while offering a scalable, practical, and flexible solution in resource-constrained settings.

## 8 Acknowledgments

This work was supported by Center for Applied Research in Artificial Intelligence (CARAI) grant funded by DAPA and ADD (UD230017TD), the National Research Foundation of Korea(NRF) grant funded by the Korea government(MSIT) (No. RS-2024-00340966 and No. RS-2024-00408003), and by the Institute for Information & Communications Technology Promotion (IITP) grant funded by the Korea government(MSIT) (No. RS-2024-00444862).

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

# 9 Supplementary Materials

This Supplementary Materials provides additional details, analyses, and results to complement the main paper. The content is organized into the following subsections:

1. **Complexity Analysis** (9.1): A detailed breakdown of the computational complexity of PruneFuse and its components.

2. **Error Analysis for PruneFuse** (9.2): An error analysis outlining proxy-target mismatch in the proposed framework.

3. **Implementation Details** (9.3): Specific details about the experimental setup, hyperparameters, and configurations used in our experiments.

4. **Details of the Fusion Process** (9.4): Specific details about the fusion process of PruneFuse.

5. **Performance Comparison with Different Datasets, Selection Metrics, and Architectures** (9.5): Results demonstrating PruneFuse's adaptability across datasets and architectures.

6. **Ablation Study of Fusion** (9.6): Analysis of the impact of the fusion process on PruneFuse's performance.

7. **Ablation Study of Knowledge Distillation in PruneFuse** (9.7): An evaluation of the role of knowledge distillation in improving performance.

8. **Comparison with SVP** (9.8): A comparison highlighting differences and improvements over the SVP baseline.

9. **Ablation Study on the Number of Selected Data Points ($k$)** (9.9): Investigation of how varying $k$ affects PruneFuse's performance.

10. **Impact of Early Stopping on Performance** (9.10): Evaluation of the utility of early stopping when integrated with PruneFuse.

11. **Performance Comparison Across Architectures and Datasets** (9.11): Additional results comparing PruneFuse's performance on various architectures and datasets.

12. **Performance at Lower Pruning Rates** (9.12): Results demonstrating PruneFuse's effectiveness at lower pruning rates.

13. **Comparison with Recent Coreset Selection Techniques** (9.13): Evaluation of PruneFuse's performance with recent coreset selection methods.

14. **Effect of Various Pruning Strategies and Criteria** (9.14): Analysis of different pruning techniques and criteria on PruneFuse's performance.

15. **Detailed Runtime Analysis of PruneFuse** (9.15): A detailed runtime analysis of PruneFuse compared to baseline methods.

Each section provides additional insights, evaluations, and experiments to further validate and explain the effectiveness of the proposed approach.

### 9.1   Complexity Analysis

Given $P$ and $N$ represent the total number of parameters in the pruned and dense model, where $P \ll N$, the computational costs can be summarized as follows:

**Initial Training on $s_0$:**

$$
\begin{aligned}
\text{PruneFuse:} \quad & O\left(|s_0| \times P \times T\right) + O\left(P \times \log P\right) \text{ one time pruning cost} \\
\text{Baseline AL:} \quad & O\left(|s_0| \times N \times T\right)
\end{aligned}
$$

**Data selection round with current labeled pool L:**

$$
\begin{aligned}
\text{PruneFuse:} \quad & O\left(|L| \times P \times T\right) + O\left(|U| \times P\right) \text{ selection} \\
\text{Baseline AL:} \quad & O\left(|L| \times N \times T\right) + O\left(|U| \times N\right) \text{ selection}
\end{aligned}
$$

**Training of the final model on the final labeled set L:**

$$
\begin{aligned}
\text{PruneFuse:} \quad & O\left(|L| \times N \times T\right) + O\left(P\right) \text{ one time fusion cost} \\
\text{Baseline AL:} \quad & O\left(|L| \times N \times T\right)
\end{aligned}
$$

**Total training complexity:**

$$
\begin{aligned}
\text{PruneFuse:} \quad & O\left(|s_0| \times P \times T\right) + O\left(P \times \log P\right) + R \times \left[O\left(|L| \times P \times T\right) + O\left(|U| \times P\right)\right] \\
& + F_{sync} * \left[O\left(|L| \times N \times T\right) + O(P) + O\left(|L| \times P \times T\right) + O\left(P \times \log P\right)\right] \\
& + O\left(|L| \times N \times T\right) + O(P) \\
\text{Baseline AL:} \quad & O\left(|s_0| \times N \times T\right) + R \times \left[O\left(|L| \times N \times T\right) + O\left(|U| \times N\right)\right] + O\left(|L| \times N \times T\right)
\end{aligned}
$$

Here $T$ represents the total number of epochs for a training round of AL which in our case is set to 181. $U$ is the whole unlabeled dataset and $R$ represents the total number of AL rounds. $F_{sync}$ represents the frequency of iterative pruning based on the fused model.

We can see that the major training costs in Active Learning (AL) arise from the repeated use of a large, dense model, which significantly increases computational expenses, especially across multiple rounds of data selection. By using a smaller surrogate (pruned model) for these rounds, as implemented in PruneFuse, the training cost and overall computation are reduced substantially. This approach leads to a more efficient and cost-effective data selection process, allowing for better resource utilization while maintaining high performance.

### 9.2   Error Decomposition for PruneFuse

This section provides the proof and additional discussion for Theorem 5.1 in the main paper.

**Setup.** Let $D = \{(x_i, y_i)\}_{i=1}^n$ be the finite pool and let $s_p \subseteq D$ be the subset of size $m$ selected at cycle $t$ by the pruned selector $\theta_p^t$. Let $\theta^t$ denote the dense target model at cycle $t$ (after fusion/fine-tuning when synchronization is performed). We analyze the representativeness gap

$$
\left| \mathbb{E}_{(x,y) \in s_p} l(x, y; \theta^t) - \mathbb{E}_{(x,y) \in D} l(x, y; \theta^t) \right|. \tag{5}
$$

**Assumption 9.1.** *The loss function $l(x, y; \theta)$ is Lipschitz continuous with respect to the model parameters $\theta$, i.e., there exists $L \geq 0$ such that*

$$
|l(x, y; \theta_1) - l(x, y; \theta_2)| \leq L \|\theta_1 - \theta_2\|. \tag{6}
$$

*This regularity condition is assumed to hold locally on the region of parameter space explored during training (e.g., between $\theta^t$ and $\theta_p^t$), and is used only to relate parameter mismatch to loss mismatch.*

**Assumption 9.2** ($\delta$-coreset under the proxy selector). *At cycle $t$, the subset $s_p \subseteq D$ returned by the acquisition rule using the pruned selector $\theta_p^t$ is a $\delta$–core-set for $D$ with respect to $\theta_p^t$ (in terms of average loss), i.e. there exists $\delta \geq 0$ such that*

$$\left| \mathbb{E}_{(x,y) \in s_p} \left[ l(x,y;\theta_p^t) \right] - \mathbb{E}_{(x,y) \in D} \left[ l(x,y;\theta_p^t) \right] \right| \leq \delta, \tag{7}$$

*with probability at least $1 - \eta$ over any randomness in the selection procedure.*

**Assumption 9.3** (Soft synchronization: selector–target proximity). *After each synchronization step, the pruned selector $\theta_p^t$ used for data acquisition is derived from the dense target model $\theta^t$ such that:*

$$\|\theta_p^t - \theta^t\| \leq \rho_t, \tag{8}$$

*for some nonnegative quantity $\rho_t$ (which may depend on $t$, the pruning ratio, and the synchronization policy). Here, we represent the pruned selector $\theta_p^t$ in the same parameter space as the dense model $\theta^t$ by embedding the pruned weights into the dense coordinates and setting the pruned coordinates to zero.*

**Theorem 9.1.** *Under Assumptions 9.1–9.3, with probability at least $1 - \eta$,*

$$\left| \mathbb{E}_{(x,y) \in s_p} l(x,y;\theta^t) - \mathbb{E}_{(x,y) \in D} l(x,y;\theta^t) \right| \leq \delta + 2L\rho_t. \tag{9}$$

**Proof.** Fix a cycle $t$. Add and subtract expectations under the pruned selector $\theta_p^t$ and apply the triangle inequality:

$$\left| \mathbb{E}_{(x,y) \in s_p} l(x,y;\theta^t) - \mathbb{E}_{(x,y) \in D} l(x,y;\theta^t) \right|$$

$$\leq \left| \mathbb{E}_{(x,y) \in s_p} l(x,y;\theta^t) - \mathbb{E}_{(x,y) \in s_p} l(x,y;\theta_p^t) \right|$$

$$+ \left| \mathbb{E}_{(x,y) \in s_p} l(x,y;\theta_p^t) - \mathbb{E}_{(x,y) \in D} l(x,y;\theta_p^t) \right|$$

$$+ \left| \mathbb{E}_{(x,y) \in D} l(x,y;\theta_p^t) - \mathbb{E}_{(x,y) \in D} l(x,y;\theta^t) \right|.$$

By Assumption 9.2, the middle term is at most $\delta$ with probability at least $1 - \eta$. By Lipschitz continuity (Assumption 9.1), for every $(x,y)$,

$$|l(x,y;\theta^t) - l(x,y;\theta_p^t)| \leq L\|\theta^t - \theta_p^t\|.$$

Taking expectation over $(x,y) \in s_p$ yields

$$\left| \mathbb{E}_{(x,y) \in s_p} l(x,y;\theta^t) - \mathbb{E}_{(x,y) \in s_p} l(x,y;\theta_p^t) \right| \leq L\|\theta^t - \theta_p^t\|.$$

Applying the same argument over $(x,y) \in D$ yields

$$\left| \mathbb{E}_{(x,y) \in D} l(x,y;\theta_p^t) - \mathbb{E}_{(x,y) \in D} l(x,y;\theta^t) \right| \leq L\|\theta^t - \theta_p^t\|.$$

Combining the three bounds, with probability at least $1 - \eta$,

$$\left| \mathbb{E}_{(x,y) \in s_p} l(x,y;\theta^t) - \mathbb{E}_{(x,y) \in D} l(x,y;\theta^t) \right| \leq \delta + 2L\|\theta^t - \theta_p^t\|.$$

Finally, Assumption 9.3 implies $\|\theta^t - \theta_p^t\| \leq \rho_t$, which gives $\delta + 2L\rho_t$. $\qquad\square$

**Interpretation.** The bound in Theorem 9.1 decomposes the representativeness error into two intuitive terms. The first term, $\delta$, captures the intrinsic selection error of the acquisition rule itself: even when selection is performed using the target model, uncertainty- or diversity-based active learning methods typically incur a nonzero approximation gap relative to the full dataset. The second term, $2L\rho_t$, is an additional penalty arising from performing selection with a proxy model rather than the target. This term quantifies the effect of proxy–target mismatch and becomes small when the proxy is well aligned with the target. (See section 9.2.1) Fusion and synchronization in PruneFuse are therefore justified as mechanisms to help control $\rho_t$, ensuring that proxy-based selection behaves similarly to selection performed directly with the target model.

### 9.2.1 Empirical Proxy-Target Alignment.

Assumption 9.3 states that the acquisition proxy remains reasonably aligned with the current target. To assess this alignment empirically, at cycle $t$ we compare the synchronized proxy $\theta_p^t$ (obtained by pruning the current dense target $\theta^t$ and fine-tuning) against a *fresh* pruned proxy $\tilde{\theta}_p^t$ trained from scratch at cycle $t$ on the same labeled set.

Let $V$ be a held-out validation set and let $\mathrm{Acc}_V(\cdot)$ denote top-1 validation accuracy. We report the target gaps

$$\Delta_t^{\mathrm{sync}} := \left|\mathrm{Acc}_V(\theta^t) - \mathrm{Acc}_V(\theta_p^t)\right|, \qquad \Delta_t^{\mathrm{fresh}} := \left|\mathrm{Acc}_V(\theta^t) - \mathrm{Acc}_V(\tilde{\theta}_p^t)\right|.$$

Our results confirm that the synchronized proxy remains closer to the target than the fresh proxy:

$$\Delta_t^{\mathrm{sync}} \le \Delta_t^{\mathrm{fresh}}. \tag{10}$$

This provides an empirical sanity check supporting the proxy–target proximity premise and is consistent with PruneFuse outperforming other proxy-based AL, where acquisition relies on a fresh proxy trained from scratch and not kept aligned with the evolving target.

Table 8: **Empirical Proxy–Target Alignment** This table summarizes performance comparison between the pruned selector and the fused dense model across label budgets, with and without synchronization on CIFAR-10 with ResNet-56 for pruning ratio $p = 0.5$. The rest of experimental setup is held same for fair comparison.

| Method | Label Budget ($b$) | | | | |
|---|---|---|---|---|---|
| | 10% | 20% | 30% | 40% | 50% |
| Data Selector ($\tilde{\theta}_P$) | 78.09 | 84.62 | 88.08 | 89.60 | 90.44 |
| Data Selector ($\theta_P$) | 79.90 | 87.35 | 90.08 | 91.28 | 91.55 |
| Target Model ($\theta$) | 81.00 | 88.54 | 91.77 | 93.12 | 93.77 |
| $\Delta_t^{\mathrm{fresh}}$ | 2.61 | 3.27 | 3.34 | 3.24 | 3.04 |
| $\Delta_t^{\mathrm{sync}}$ | **1.10** | **1.19** | **1.69** | **1.84** | **2.22** |

Table 8 reports accuracies of the pruned selector and the fused dense model, along with their gap, across label budgets. It can be noted that the synchronization reduces the proxy–target gap, providing empirical evidence that synchronization helps maintain proxy–target alignment as assumed in Assumption 9.3.

## 9.3 Implementation Details

We used ResNet-50, ResNet-56, ResNet-110, ResNet-164, Wide-ResNet, VDCNN, and Vision transformers architectures in our experiments. We pruned these architectures using the Torch-Pruning library (Fang et al., 2023) for different pruning ratios $p = 0.5, 0.6, 0.7,$ and $0.8$ to get the pruned architectures. For CIFAR-10 and CIFAR-100, the models were trained for 181 epochs, with an epoch schedule of [1, 90, 45, 45], and corresponding learning rates of [0.01, 0.1, 0.01, 0.001], using a momentum of 0.9 and weight decay of 0.0005. For TinyImageNet-200 and ImageNet-1K, the models were trained over an epoch schedule of [1, 1, 1, 1, 1, 25, 30, 20, 20], with learning rates of [0.0167, 0.0333, 0.05, 0.0667, 0.0833, 0.1, 0.01, 0.001, 0.0001], a momentum of 0.9, and weight decay of 0.0001. We use the mini-batch of 128 for CIFAR-10 and CIFAR-100 and 256 for TinyImageNet-200 and ImageNet-1K. We also extend our experiments to text datasets: Amazon Review Polarity and Full (Zhang & LeCun, 2015; Zhang et al., 2015). Amazon Review Polarity has 3.6 million reviews split evenly between positive and negative ratings, with an additional 400,000 reviews for testing. Amazon Review Full has 3 million reviews split evenly between the 5 stars with an additional 650,000 reviews for testing. For Amazon Review Polarity and Full, the models were trained over an epoch schedule of [3, 3, 3, 3, 3], with learning rates of [0.01, 0.005, 0.0025, 0.00125, 0.000625], a momentum of 0.9, weight decay of 0.0001, and a mini-batch size of 128. For all the experiments SGD is used as an optimizer. We set the knowledge distillation coefficient $\lambda$ to 0.3. We took Active Learning (AL) as a baseline for the proposed technique and initially, we started by randomly selecting 2% of the data. For the first round, we added 8% from the unlabeled set, then 10% in each subsequent round, until reaching the label budget, $b$. After each round, we retrained the models from scratch, as described in the methodology. All experiments are carried out independently 3 times and then the average is reported.

### 9.4 Additional Details of the Fusion Process

*Convolutional layers.* Let $W^{(j)} \in \mathbb{R}^{C_{\text{out}}^{(j)} \times C_{\text{in}}^{(j)} \times k_h \times k_w}$ denote the dense weights of layer $j$, and $W_{p^*}^{(j)} \in \mathbb{R}^{|I_j| \times |I_{j-1}| \times k_h \times k_w}$ the trained pruned weights. The weight-aligned fusion copies the trained sub-tensor into the matching coordinates of the dense tensor and keeps all remaining entries at their initial values:

$$W_F^{(j)}[I_j, I_{j-1}, :, :] \leftarrow W_{p^*}^{(j)},$$
$$W_F^{(j)}[I_j, \overline{I}_{j-1}, :, :] \leftarrow W_{\text{init}}^{(j)}[I_j, \overline{I}_{j-1}, :, :],$$
$$W_F^{(j)}[\overline{I}_j, :, :, :] \leftarrow W_{\text{init}}^{(j)}[\overline{I}_j, :, :, :],$$

with the same coordinate replacement for biases (if present) and normalization parameters $(\gamma, \beta, \mu, \sigma^2)$ on the indices $I_j$.

*Linear layers.* For $W^{(j)} \in \mathbb{R}^{C_{\text{out}}^{(j)} \times C_{\text{in}}^{(j)}}$, the output dimension (e.g., number of classes) is unchanged; fusion aligns on the input channels:

$$W_F^{(j)}[:, I_{j-1}] \leftarrow W_{p^*}^{(j)}, \qquad W_F^{(j)}[:, \overline{I}_{j-1}] \leftarrow W_{\text{init}}^{(j)}[:, \overline{I}_{j-1}].$$

### 9.5 Performance Comparison with different Datasets, Selection Metrics, and Architectures

To comprehensively evaluate the effectiveness of PruneFuse, we conducted additional experiments comparing its performance with baseline using other data selection metrics such as Least Confidence, Entropy, and Greedy k-centers. Results are shown in Tables 9, 10, 11 and 12 for various architectures and labeling budgets. In all cases, our results demonstrate that PruneFuse mostly outperforms the baseline using these traditional metrics across various datasets and model architectures, highlighting the robustness of PruneFuse in selecting the most informative samples efficiently.

We further performed experiments on ViT, MobileNet for Vision task in Table 13, 14 and VDCNN for NLP tasks in Table 15, 16, to underscore PruneFuse's consistent efficiency and robust accuracy across different architectures and domains. Moreover, we demonstrated that PruneFuse does not degrade performance on OOD datasets in Table 17, reinforcing PruneFuse's stability.

Table 9: **Performance Comparison** of Baseline and PruneFuse on CIFAR-10 and CIFAR-100 with ResNet-56 architecture. This table summarizes the test accuracy of final models (original in case of AL and Fused in PruneFuse) for various pruning ratios ($p$), labeling budgets ($b$), and data selection metrics.

| Method | Selection Metric | Label Budget ($b$) | | | | |
|---|---|---|---|---|---|---|
| | | 10% | 20% | 30% | 40% | 50% |
| Baseline AL | Least Conf | $80.53 \pm 0.20$ | $87.74 \pm 0.15$ | $90.85 \pm 0.11$ | $92.24 \pm 0.16$ | $93.00 \pm 0.11$ |
| | Entropy | $80.14 \pm 0.41$ | $87.63 \pm 0.10$ | $90.80 \pm 0.36$ | $92.51 \pm 0.34$ | $92.98 \pm 0.03$ |
| | Random | $78.55 \pm 0.38$ | $85.26 \pm 0.21$ | $88.13 \pm 0.35$ | $89.81 \pm 0.15$ | $91.20 \pm 0.05$ |
| | Greedy k | $79.63 \pm 0.83$ | $86.46 \pm 0.27$ | $90.09 \pm 0.20$ | $91.9 \pm 0.08$ | $92.80 \pm 0.08$ |
| PruneFuse $p = 0.5$ | Least Conf | $80.92 \pm 0.41$ | $88.35 \pm 0.33$ | $91.44 \pm 0.15$ | $92.77 \pm 0.03$ | $93.65 \pm 0.14$ |
| | Entropy | $81.08 \pm 0.16$ | $88.74 \pm 0.10$ | $91.33 \pm 0.04$ | $92.78 \pm 0.04$ | $93.48 \pm 0.04$ |
| | Random | $80.43 \pm 0.27$ | $86.28 \pm 0.37$ | $88.75 \pm 0.17$ | $90.36 \pm 0.02$ | $91.42 \pm 0.12$ |
| | Greedy k | $79.85 \pm 0.68$ | $86.96 \pm 0.38$ | $90.20 \pm 0.16$ | $91.82 \pm 0.14$ | $92.89 \pm 0.14$ |
| PruneFuse $p = 0.6$ | Least Conf | $80.58 \pm 0.33$ | $87.79 \pm 0.20$ | $90.94 \pm 0.13$ | $92.58 \pm 0.31$ | $93.08 \pm 0.42$ |
| | Entropy | $80.96 \pm 0.16$ | $87.89 \pm 0.45$ | $91.22 \pm 0.28$ | $92.56 \pm 0.19$ | $93.19 \pm 0.26$ |
| | Random | $79.19 \pm 0.57$ | $85.65 \pm 0.29$ | $88.27 \pm 0.18$ | $90.13 \pm 0.24$ | $91.01 \pm 0.28$ |
| | Greedy k | $79.54 \pm 0.48$ | $86.16 \pm 0.60$ | $89.50 \pm 0.29$ | $91.35 \pm 0.06$ | $92.39 \pm 0.22$ |
| PruneFuse $p = 0.7$ | Least Conf | $80.19 \pm 0.45$ | $87.88 \pm 0.05$ | $90.70 \pm 0.21$ | $92.44 \pm 0.24$ | $93.40 \pm 0.11$ |
| | Entropy | $79.73 \pm 0.87$ | $87.85 \pm 0.25$ | $90.94 \pm 0.29$ | $92.41 \pm 0.23$ | $93.39 \pm 0.20$ |
| | Random | $78.76 \pm 0.23$ | $85.50 \pm 0.11$ | $88.31 \pm 0.19$ | $89.94 \pm 0.24$ | $90.87 \pm 0.17$ |
| | Greedy k | $78.93 \pm 0.15$ | $85.85 \pm 0.41$ | $88.96 \pm 0.07$ | $90.93 \pm 0.19$ | $92.23 \pm 0.08$ |
| PruneFuse $p = 0.8$ | Least Conf | $80.11 \pm 0.28$ | $87.58 \pm 0.14$ | $90.50 \pm 0.08$ | $92.42 \pm 0.41$ | $93.32 \pm 0.14$ |
| | Entropy | $79.83 \pm 1.13$ | $87.50 \pm 0.54$ | $90.52 \pm 0.24$ | $92.24 \pm 0.13$ | $93.15 \pm 0.10$ |
| | Random | $78.77 \pm 0.66$ | $85.64 \pm 0.13$ | $88.45 \pm 0.33$ | $89.88 \pm 0.14$ | $91.21 \pm 0.43$ |
| | Greedy k | $78.23 \pm 0.37$ | $85.59 \pm 0.25$ | $88.60 \pm 0.19$ | $90.11 \pm 0.11$ | $91.31 \pm 0.08$ |

(a) CIFAR-10 using ResNet-56 architecture.

| Method | Selection Metric | Label Budget ($b$) | | | | |
|---|---|---|---|---|---|---|
| | | 10% | 20% | 30% | 40% | 50% |
| Baseline AL | Least Conf | $35.99 \pm 0.80$ | $52.99 \pm 0.56$ | $59.29 \pm 0.46$ | $63.68 \pm 0.53$ | $66.72 \pm 0.33$ |
| | Entropy | $37.57 \pm 0.51$ | $52.64 \pm 0.76$ | $58.87 \pm 0.38$ | $63.97 \pm 0.17$ | $66.78 \pm 0.27$ |
| | Random | $37.06 \pm 0.64$ | $51.62 \pm 0.21$ | $58.77 \pm 0.65$ | $62.05 \pm 0.02$ | $64.63 \pm 0.16$ |
| | Greedy k | $38.28 \pm 1.11$ | $52.43 \pm 0.24$ | $58.96 \pm 0.16$ | $63.56 \pm 0.30$ | $66.30 \pm 0.31$ |
| PruneFuse $p = 0.5$ | Least Conf | $40.26 \pm 0.95$ | $53.90 \pm 1.06$ | $60.80 \pm 0.44$ | $64.98 \pm 0.4$ | $67.87 \pm 0.17$ |
| | Entropy | $38.59 \pm 1.67$ | $54.01 \pm 1.17$ | $60.52 \pm 0.19$ | $64.83 \pm 0.27$ | $67.67 \pm 0.33$ |
| | Random | $39.43 \pm 0.99$ | $54.60 \pm 0.64$ | $60.13 \pm 0.96$ | $63.91 \pm 0.39$ | $66.02 \pm 0.3$ |
| | Greedy k | $39.83 \pm 2.44$ | $54.35 \pm 0.41$ | $60.40 \pm 0.23$ | $64.22 \pm 0.25$ | $66.89 \pm 0.16$ |
| PruneFuse $p = 0.6$ | Least Conf | $37.82 \pm 0.83$ | $52.65 \pm 0.4$ | $60.08 \pm 0.22$ | $63.7 \pm 0.25$ | $66.89 \pm 0.46$ |
| | Entropy | $38.01 \pm 0.79$ | $51.91 \pm 0.56$ | $59.18 \pm 0.31$ | $63.53 \pm 0.25$ | $66.88 \pm 0.18$ |
| | Random | $38.27 \pm 0.81$ | $52.85 \pm 1.22$ | $58.68 \pm 0.68$ | $62.28 \pm 0.22$ | $65.2 \pm 0.48$ |
| | Greedy k | $38.44 \pm 0.98$ | $52.85 \pm 0.74$ | $59.36 \pm 0.57$ | $63.36 \pm 0.75$ | $66.12 \pm 0.38$ |
| PruneFuse $p = 0.7$ | Least Conf | $36.76 \pm 0.63$ | $52.15 \pm 0.53$ | $59.33 \pm 0.17$ | $63.65 \pm 0.36$ | $66.84 \pm 0.43$ |
| | Entropy | $36.95 \pm 1.03$ | $50.64 \pm 0.33$ | $58.45 \pm 0.36$ | $62.27 \pm 0.27$ | $65.88 \pm 0.28$ |
| | Random | $37.30 \pm 1.24$ | $51.66 \pm 0.21$ | $58.79 \pm 0.13$ | $62.67 \pm 0.29$ | $65.08 \pm 0.08$ |
| | Greedy k | $38.88 \pm 2.18$ | $52.02 \pm 0.77$ | $58.66 \pm 0.19$ | $61.39 \pm 0.11$ | $65.28 \pm 0.65$ |
| PruneFuse $p = 0.8$ | Least Conf | $36.49 \pm 0.20$ | $50.98 \pm 0.54$ | $58.53 \pm 0.50$ | $62.87 \pm 0.13$ | $65.85 \pm 0.32$ |
| | Entropy | $36.02 \pm 1.30$ | $51.23 \pm 0.23$ | $57.44 \pm 0.11$ | $62.65 \pm 0.46$ | $65.76 \pm 0.30$ |
| | Random | $37.37 \pm 0.85$ | $52.06 \pm 0.47$ | $58.19 \pm 0.30$ | $62.19 \pm 0.45$ | $64.77 \pm 0.29$ |
| | Greedy k | $37.04 \pm 0.09$ | $49.84 \pm 0.49$ | $56.13 \pm 0.20$ | $60.24 \pm 0.42$ | $62.92 \pm 0.44$ |

(b) CIFAR-100 using ResNet-56 architecture.

Table 10: **Performance Comparison** of Baseline and PruneFuse on CIFAR-10 and CIFAR-100 with ResNet-110 architecture. This table summarizes the test accuracy of final models (original in case of AL and Fused in PruneFuse) for various pruning ratios ($p$), labeling budgets ($b$), and data selection metrics.

| Method | Selection Metric | Label Budget ($b$) | | | | |
|---|---|---|---|---|---|---|
| | | 10% | 20% | 30% | 40% | 50% |
| Baseline AL | Least Conf. | 80.74 ± 0.04 | 87.80 ± 0.09 | 91.50 ± 0.09 | 93.19 ± 0.14 | 93.68 ± 0.17 |
| | Entropy | 79.81 ± 0.18 | 88.46 ± 0.30 | 91.30 ± 0.15 | 92.83 ±0.30 | 93.47 ± 0.31 |
| | Random | 79.99 ± 0.10 | 85.63 ± 0.03 | 88.07 ± 0.31 | 90.40 ± 0.42 | 91.42 ± 0.26 |
| | Greedy k | 78.69 ± 0.58 | 87.46 ±0.20 | 90.72 ± 0.14 | 92.55 ±0.14 | 93.44 ± 0.07 |
| PruneFuse $p = 0.5$ | Least Conf. | 81.24 ± 0.43 | 88.70 ± 0.15 | 92.02 ± 0.10 | 93.32 ± 0.13 | 94.07 ± 0.06 |
| | Entropy | 81.45 ± 0.39 | 88.90 ± 0.11 | 92.13 ± 0.15 | 93.49 ± 0.16 | 94.07 ± 0.05 |
| | Random | 80.08 ± 0.86 | 86.52 ± 0.14 | 89.48 ± 0.16 | 90.82 ± 0.21 | 91.79 ± 0.04 |
| | Greedy k | 80.40 ± 0.09 | 87.77 ± 0.13 | 90.74 ± 0.09 | 92.48 ± 0.22 | 93.53 ± 0.22 |
| PruneFuse $p = 0.6$ | Least Conf. | 81.12 ± 0.34 | 88.33 ± 0.31 | 91.57 ± 0.03 | 93.25 ± 0.21 | 93.90 ± 0.17 |
| | Entropy | 80.02 ± 0.41 | 88.49 ± 0.18 | 91.51 ± 0.14 | 93.03 ± 0.11 | 93.94 ± 0.12 |
| | Random | 78.55 ± 0.42 | 85.94 ± 0.34 | 88.77 ± 0.10 | 90.66 ± 0.20 | 92.02 ± 0.03 |
| | Greedy k | 79.44 ± 0.28 | 87.05 ± 0.63 | 90.30 ± 0.15 | 92.15 ± 0.12 | 93.22 ± 0.04 |
| PruneFuse $p = 0.7$ | Least Conf. | 79.93 ± 0.06 | 88.04 ± 0.23 | 91.51 ± 0.34 | 92.90 ± 0.02 | 93.82 ± 0.09 |
| | Entropy | 80.16 ± 0.27 | 87.78 ± 0.52 | 91.21 ± 0.13 | 92.99 ± 0.13 | 93.81 ± 0.12 |
| | Random | 79.41 ± 0.36 | 86.14 ± 0.44 | 88.86 ± 0.11 | 90.35 ± 0.08 | 91.35 ± 0.24 |
| | Greedy k | 78.58 ± 0.91 | 86.37 ± 0.36 | 89.70 ± 0.33 | 91.71 ± 0.18 | 92.97 ± 0.10 |
| PruneFuse $p = 0.8$ | Least Conf. | 80.34 ± 0.39 | 88.00 ± 0.13 | 91.22 ± 0.07 | 92.89 ± 0.23 | 93.80 ± 0.23 |
| | Entropy | 79.61 ± 0.35 | 88.12 ± 0.00 | 90.94 ± 0.13 | 92.76 ± 0.14 | 93.54 ± 0.24 |
| | Random | 78.94 ± 0.49 | 86.20 ± 0.10 | 89.11 ± 0.34 | 90.50 ± 0.22 | 91.42 ± 0.23 |
| | Greedy k | 78.41 ± 0.76 | 85.90 ± 0.73 | 89.57 ± 0.51 | 91.38 ± 0.32 | 92.21± 0.22 |

(a) CIFAR-10 using ResNet-110 architecture.

| Method | Selection Metric | Label Budget ($b$) | | | | |
|---|---|---|---|---|---|---|
| | | 10% | 20% | 30% | 40% | 50% |
| Baseline AL | Least Conf. | 38.61 ±0.32 | 54.47 ±0.56 | 61.46 ±0.25 | 65.96 ±0.48 | 68.91 ± 0.40 |
| | Entropy | 38.00 ± 0.99 | 54.71 ±0.83 | 60.82 ±0.15 | 66.19 ± 0.31 | 68.79 ± 0.50 |
| | Random | 37.88 ± 1.03 | 52.84 ±0.11 | 59.41 ±0.34 | 64.11 ± 0.11 | 67.22 ± 0.36 |
| | Greedy k | 37.41 ± 0.98 | 53.86 ±0.55 | 61.44 ±0.26 | 65.73 ± 0.50 | 68.17 ± 0.46 |
| PruneFuse $p = 0.5$ | Least Conf. | 41.42 ± 0.51 | 55.91 ± 0.36 | 62.43 ± 0.32 | 66.95 ± 0.20 | 69.79 ± 0.26 |
| | Entropy | 40.83 ± 0.59 | 56.29 ± 0.83 | 62.62 ± 0.45 | 66.91 ± 0.02 | 69.96 ± 0.39 |
| | Random | 40.36 ± 0.74 | 55.48 ± 0.25 | 61.14 ± 0.68 | 65.03 ± 0.42 | 67.85 ± 0.53 |
| | Greedy k | 41.22 ± 0.46 | 55.70 ± 0.54 | 62.27 ± 0.02 | 66.20 ± 0.14 | 68.86 ± 0.14 |
| PruneFuse $p = 0.6$ | Least Conf. | 38.52 ± 1.49 | 54.90 ± 0.32 | 61.50 ± 0.77 | 66.14 ± 0.68 | 69.03 ± 0.24 |
| | Entropy | 38.78 ± 1.35 | 53.13 ± 0.30 | 61.42 ± 0.14 | 65.62 ± 0.43 | 68.89 ± 0.09 |
| | Random | 40.24 ± 0.90 | 53.38 ± 0.68 | 59.93 ± 0.12 | 64.70 ± 0.15 | 66.62 ± 0.24 |
| | Greedy k | 39.99 ± 1.56 | 54.91 ± 2.23 | 61.04 ± 0.25 | 64.69 ± 0.63 | 67.60 ± 0.08 |
| PruneFuse $p = 0.7$ | Least Conf. | 37.83 ± 1.02 | 53.08 ± 0.25 | 61.41 ± 0.21 | 65.77 ± 0.43 | 68.03 ± 0.14 |
| | Entropy | 36.53 ± 0.97 | 52.97 ± 0.76 | 59.82 ± 0.63 | 64.97 ± 0.13 | 68.64 ± 0.54 |
| | Random | 39.46 ± 0.59 | 52.89 ± 0.77 | 59.92 ± 0.55 | 63.69 ± 0.25 | 66.30 ± 0.15 |
| | Greedy k | 40.44 ± 0.13 | 52.56 ± 0.28 | 59.83 ± 0.45 | 64.50 ± 0.29 | 66.99 ± 0.50 |
| PruneFuse $p = 0.8$ | Least Conf. | 38.33 ± 0.58 | 52.89 ± 0.49 | 60.08 ± 0.32 | 65.12 ± 0.60 | 68.06 ± 0.56 |
| | Entropy | 35.34 ± 0.98 | 51.88 ± 0.74 | 59.80 ± 0.82 | 64.58 ± 0.43 | 68.02 ± 0.17 |
| | Random | 38.22 ± 0.39 | 53.37 ± 0.72 | 59.84 ± 0.43 | 64.31 ± 0.33 | 67.23 ± 0.25 |
| | Greedy k | 37.72 ± 0.70 | 50.55 ± 1.79 | 57.39 ± 0.93 | 61.79 ± 0.53 | 65.21 ± 0.24 |

(b) CIFAR-100 using ResNet-110 architecture.

Table 11: **Performance Comparison** of Baseline and PruneFuse on CIFAR-10 and CIFAR-100 with ResNet-164 architecture. This table summarizes the test accuracy of final models (original in case of AL and Fused in PruneFuse) for various pruning ratios ($p$), labeling budgets ($b$), and data selection metrics.

| Method | Selection Metric | Label Budget ($b$) | | | | |
|---|---|---|---|---|---|---|
| | | 10% | 20% | 30% | 40% | 50% |
| Baseline AL | Least Conf. | $81.15 \pm 0.52$ | $89.4 \pm 0.27$ | $92.72 \pm 0.10$ | $94.09 \pm 0.14$ | $94.63 \pm 0.18$ |
| | Entropy | $80.99 \pm 0.44$ | $89.54 \pm 0.18$ | $92.45 \pm 0.16$ | $94.06 \pm 0.05$ | $94.49 \pm 0.09$ |
| | Random | $80.27 \pm 0.18$ | $87.00 \pm 0.08$ | $89.94 \pm 0.13$ | $91.57 \pm 0.09$ | $92.78 \pm 0.04$ |
| | Greedy k | $80.02 \pm 0.42$ | $88.33 \pm 0.47$ | $91.76 \pm 0.24$ | $93.39 \pm 0.22$ | $94.40 \pm 0.18$ |
| PruneFuse $p = 0.5$ | Least Conf. | $83.03 \pm 0.09$ | $90.30 \pm 0.06$ | $93.00 \pm 0.15$ | $94.41 \pm 0.08$ | $94.63 \pm 0.13$ |
| | Entropy | $82.64 \pm 0.22$ | $89.88 \pm 0.27$ | $93.08 \pm 0.25$ | $94.32 \pm 0.12$ | $94.90 \pm 0.13$ |
| | Random | $81.52 \pm 0.54$ | $87.84 \pm 0.15$ | $90.14 \pm 0.08$ | $91.94 \pm 0.18$ | $92.81 \pm 0.12$ |
| | Greedy k | $81.70 \pm 0.13$ | $88.75 \pm 0.33$ | $91.92 \pm 0.07$ | $93.64 \pm 0.04$ | $94.22 \pm 0.09$ |
| PruneFuse $p = 0.6$ | Least Conf. | $82.86 \pm 0.38$ | $90.22 \pm 0.18$ | $93.05 \pm 0.10$ | $94.27 \pm 0.06$ | $94.66 \pm 0.08$ |
| | Entropy | $82.23 \pm 0.39$ | $90.18 \pm 0.11$ | $92.91 \pm 0.15$ | $94.28 \pm 0.14$ | $94.66 \pm 0.14$ |
| | Random | $81.14 \pm 0.26$ | $87.51 \pm 0.26$ | $90.05 \pm 0.20$ | $91.82 \pm 0.22$ | $92.43 \pm 0.20$ |
| | Greedy k | $81.11 \pm 0.10$ | $88.41 \pm 0.18$ | $91.66 \pm 0.18$ | $92.94 \pm 0.12$ | $94.17 \pm 0.02$ |
| PruneFuse $p = 0.7$ | Least Conf. | $82.76 \pm 0.29$ | $89.89 \pm 0.17$ | $92.83 \pm 0.08$ | $94.10 \pm 0.08$ | $94.69 \pm 0.13$ |
| | Entropy | $82.59 \pm 0.69$ | $89.81 \pm 0.24$ | $92.77 \pm 0.07$ | $94.20 \pm 0.20$ | $94.74 \pm 0.02$ |
| | Random | $80.88 \pm 0.38$ | $87.54 \pm 0.26$ | $90.09 \pm 0.08$ | $91.57 \pm 0.26$ | $92.64 \pm 0.10$ |
| | Greedy k | $81.68 \pm 0.40$ | $88.36 \pm 0.56$ | $91.64 \pm 0.40$ | $93.02 \pm 0.42$ | $93.97 \pm 0.51$ |
| PruneFuse $p = 0.8$ | Least Conf. | $82.66 \pm 0.09$ | $89.78 \pm 0.27$ | $92.64 \pm 0.14$ | $94.08 \pm 0.10$ | $94.69 \pm 0.17$ |
| | Entropy | $82.01 \pm 0.88$ | $89.77 \pm 0.44$ | $92.65 \pm 0.09$ | $94.02 \pm 0.17$ | $94.60 \pm 0.18$ |
| | Random | $80.73 \pm 0.49$ | $87.43 \pm 0.44$ | $90.08 \pm 0.12$ | $91.40 \pm 0.07$ | $92.53 \pm 0.18$ |
| | Greedy k | $79.66 \pm 0.60$ | $87.56 \pm 0.12$ | $90.79 \pm 0.07$ | $92.30 \pm 0.12$ | $93.17 \pm 0.14$ |

(a) CIFAR-10 using ResNet-164 architecture.

| Method | Selection Metric | Label Budget ($b$) | | | | |
|---|---|---|---|---|---|---|
| | | 10% | 20% | 30% | 40% | 50% |
| Baseline AL | Least Conf | $38.41 \pm 0.73$ | $51.39 \pm 0.30$ | $65.53 \pm 0.31$ | $70.07 \pm 0.17$ | $73.05 \pm 0.11$ |
| | Entropy | $36.65 \pm 0.76$ | $57.58 \pm 0.63$ | $64.98 \pm 0.30$ | $69.99 \pm 0.17$ | $72.90 \pm 0.15$ |
| | Random | $39.31 \pm 1.22$ | $57.53 \pm 0.26$ | $63.84 \pm 0.14$ | $67.75 \pm 0.14$ | $70.79 \pm 0.07$ |
| | Greedy k | $39.76 \pm 0.58$ | $57.40 \pm 0.20$ | $65.20 \pm 0.31$ | $69.25 \pm 0.40$ | $72.91 \pm 0.29$ |
| PruneFuse $p = 0.5$ | Least Conf | $42.88 \pm 1.11$ | $59.31 \pm 0.70$ | $66.95 \pm 0.30$ | $71.45 \pm 0.42$ | $74.32 \pm 0.58$ |
| | Entropy | $42.99 \pm 0.18$ | $59.32 \pm 1.25$ | $66.83 \pm 0.29$ | $71.18 \pm 0.40$ | $74.43 \pm 0.34$ |
| | Random | $43.72 \pm 1.05$ | $58.58 \pm 0.61$ | $64.93 \pm 0.43$ | $68.75 \pm 0.57$ | $71.63 \pm 0.40$ |
| | Greedy k | $43.61 \pm 0.91$ | $58.38 \pm 0.24$ | $66.04 \pm 0.21$ | $69.83 \pm 0.16$ | $73.10 \pm 0.39$ |
| PruneFuse $p = 0.6$ | Least Conf | $41.86 \pm 0.70$ | $58.97 \pm 0.50$ | $66.61 \pm 0.39$ | $70.59 \pm 0.11$ | $73.60 \pm 0.10$ |
| | Entropy | $42.43 \pm 0.95$ | $58.74 \pm 0.80$ | $65.97 \pm 0.39$ | $70.90 \pm 0.48$ | $73.70 \pm 0.09$ |
| | Random | $42.53 \pm 0.46$ | $58.33 \pm 0.42$ | $65.00 \pm 0.26$ | $68.55 \pm 0.30$ | $71.46 \pm 0.32$ |
| | Greedy k | $42.71 \pm 0.91$ | $58.41 \pm 0.18$ | $65.43 \pm 0.69$ | $69.57 \pm 0.14$ | $72.49 \pm 0.25$ |
| PruneFuse $p = 0.7$ | Least Conf | $42.00 \pm 0.20$ | $57.08 \pm 0.36$ | $66.41 \pm 0.30$ | $70.68 \pm 0.29$ | $73.63 \pm 0.29$ |
| | Entropy | $41.01 \pm 1.66$ | $57.45 \pm 0.50$ | $65.99 \pm 0.10$ | $70.07 \pm 0.54$ | $73.45 \pm 0.04$ |
| | Random | $42.76 \pm 1.00$ | $57.31 \pm 0.07$ | $64.12 \pm 0.57$ | $68.07 \pm 0.24$ | $70.88 \pm 0.25$ |
| | Greedy k | $42.42 \pm 0.32$ | $57.58 \pm 0.52$ | $65.18 \pm 0.51$ | $68.55 \pm 0.10$ | $71.89 \pm 0.16$ |
| PruneFuse $p = 0.8$ | Least Conf | $41.19 \pm 1.07$ | $57.98 \pm 0.70$ | $65.22 \pm 0.44$ | $70.38 \pm 0.22$ | $73.17 \pm 0.26$ |
| | Entropy | $39.78 \pm 1.16$ | $57.30 \pm 0.41$ | $65.19 \pm 0.63$ | $69.40 \pm 0.34$ | $72.82 \pm 0.03$ |
| | Random | $42.08 \pm 1.55$ | $57.23 \pm 0.47$ | $64.05 \pm 0.40$ | $67.85 \pm 0.19$ | $70.62 \pm 0.06$ |
| | Greedy k | $42.20 \pm 1.21$ | $57.42 \pm 0.50$ | $64.53 \pm 0.21$ | $68.01 \pm 0.40$ | $71.29 \pm 0.14$ |

(b) CIFAR-100 using ResNet-164 architecture.

Table 12: **Performance Comparison** of Baseline and PruneFuse on **Tiny ImageNet-200 with ResNet-50 architecture**, including test accuracy and corresponding standard deviations. This table summarizes the test accuracy of final models (original in case of AL and Fused in PruneFuse) for various pruning ratios ($p$) and labeling budgets ($b$).

| Method | Label Budget ($b$) | | | | |
|---|---|---|---|---|---|
| | 10% | 20% | 30% | 40% | 50% |
| Baseline ($AL$) | $14.86 \pm 0.11$ | $33.62 \pm 0.52$ | $43.96 \pm 0.22$ | $49.86 \pm 0.56$ | $54.65 \pm 0.38$ |
| PruneFuse ($p = 0.5$) | $18.71 \pm 0.21$ | $39.70 \pm 0.31$ | $47.41 \pm 0.20$ | $51.84 \pm 0.10$ | $55.89 \pm 1.21$ |
| PruneFuse ($p = 0.6$) | $19.25 \pm 0.72$ | $38.84 \pm 0.70$ | $47.02 \pm 0.30$ | $52.09 \pm 0.29$ | $55.29 \pm 0.28$ |
| PruneFuse ($p = 0.7$) | $18.32 \pm 0.95$ | $39.24 \pm 0.75$ | $46.45 \pm 0.58$ | $52.02 \pm 0.65$ | $55.63 \pm 0.55$ |
| PruneFuse ($p = 0.8$) | $18.34 \pm 0.93$ | $37.86 \pm 0.42$ | $47.15 \pm 0.31$ | $51.77 \pm 0.40$ | $55.18 \pm 0.50$ |

Table 13: **Performance Comparison** of Baseline and PruneFuse on CIFAR-10 with **MobileNetV2 Architecture**. This table summarizes the test accuracy of final models (original in case of AL and Fused in PruneFuse) for various pruning ratios ($p$) and labeling budgets ($b$).

| Method | Label Budget ($b$) | | | | |
|---|---|---|---|---|---|
| | 10% | 20% | 30% | 40% | 50% |
| Baseline ($AL$) | 81.63 | 89.47 | 91.25 | 91.54 | 91.85 |
| PruneFuse ($p = 0.5$) | **84.49** | **90.07** | **92.63** | **93.49** | **93.55** |
| PruneFuse ($p = 0.6$) | **84.16** | **90.22** | **92.56** | **93.34** | **93.43** |
| PruneFuse ($p = 0.7$) | **84.10** | **90.21** | **92.46** | **93.29** | **93.22** |

Table 14: **Performance Comparison** of Baseline and PruneFuse on CIFAR-10 with **Vision Transformers (ViT)**. This table summarizes the test accuracy of final models (original in case of AL and Fused in PruneFuse) for various pruning ratios ($p$) and labeling budgets ($b$).

| Method | Label Budget ($b$) | | | | |
|---|---|---|---|---|---|
| | 10% | 20% | 30% | 40% | 50% |
| Baseline ($AL$) | 53.63 | 65.62 | 71.09 | 76.86 | 80.59 |
| PruneFuse ($p = 0.5$) | **59.32** | **71.63** | **75.61** | **81.50** | **83.64** |
| PruneFuse ($p = 0.6$) | **57.80** | **70.45** | **75.22** | **80.22** | **82.77** |
| PruneFuse ($p = 0.7$) | **56.17** | **69.25** | **73.87** | **79.47** | **82.15** |

Table 15: **Performance Comparison** of Baseline and PruneFuse on **Amazon Review Polarity Dataset with VDCNN Architecture**. This table summarizes the test accuracy of final models (original in case of AL and Fused in PruneFuse) for various pruning ratios ($p$) and labeling budgets ($b$).

| Method | Label Budget ($b$) | | | | |
|---|---|---|---|---|---|
| | 10% | 20% | 30% | 40% | 50% |
| Baseline ($AL$) | 94.13 | 95.06 | 95.54 | 95.73 | 95.71 |
| PruneFuse ($p = 0.5$) | **94.66** | **95.45** | **95.71** | **95.87** | **95.87** |
| PruneFuse ($p = 0.6$) | **94.62** | **95.38** | **95.69** | **95.82** | **95.88** |
| PruneFuse ($p = 0.7$) | **94.47** | **95.43** | **95.71** | **95.83** | **95.84** |
| PruneFuse ($p = 0.8$) | **94.33** | **95.37** | **95.63** | **95.79** | **95.85** |

## 9.6 Ablation Study of Fusion

The fusion process is a critical component of the PruneFuse methodology, designed to integrate the knowledge gained by the pruned model into the original network. To isolate the effect of fusion on optimization dynamics, we evaluate *fusion vs. no fusion* and also against knowledge distillation under a controlled, matched training

Table 16: **Performance Comparison** of Baseline and PruneFuse on **Amazon Review Full Dataset with VDCNN Architecture**. This table summarizes the test accuracy of final models (original in case of AL and Fused in PruneFuse) for various pruning ratios ($p$) and labeling budgets ($b$).

| Method | Label Budget ($b$) | | | | |
|---|---|---|---|---|---|
| | 10% | 20% | 30% | 40% | 50% |
| Baseline ($AL$) | 58.46 | 60.65 | 61.50 | 62.20 | 62.43 |
| PruneFuse ($p = 0.5$) | **59.45** | **61.28** | **62.02** | **62.66** | **62.84** |
| PruneFuse ($p = 0.6$) | **59.28** | **61.14** | **62.08** | **62.62** | **62.81** |
| PruneFuse ($p = 0.7$) | **59.42** | **61.05** | **61.98** | **62.48** | **62.85** |
| PruneFuse ($p = 0.8$) | **59.24** | **61.05** | **61.94** | **62.45** | **62.77** |

Table 17: **Results of CIFAR-10 (in-distribution, ID) and CIFAR-10-C (OOD corruptions) using a ResNet-56 backbone.**

| Method | Label Budget ($b$) | | | | |
|---|---|---|---|---|---|
| | 10% | 20% | 30% | 40% | 50% |
| Baseline ($AL$) (ID) | 48.67 | 58.95 | 65.23 | 72.30 | 73.23 |
| PruneFuse ($p = 0.5$) (ID) | **51.20** | **64.68** | **67.52** | **73.28** | **77.71** |
| Baseline ($AL$) (OOD) | 42.95 | 48.56 | 53.62 | 57.27 | 58.32 |
| PruneFuse ($p = 0.5$) (OOD) | **44.58** | **52.44** | **54.60** | **58.36** | **63.37** |

Table 18: **Initialization schemes for Fused Model.** Performance comparison of initializing the remaining weights of the Fused model after fusion with pruned model via 1) retaining weights from first initialization during pruned model initialization, 2) zero weights and 3) random re-initialization (PruneFuse) on ResNet-56 with CIFAR-10 dataset.

| PruneFuse Method | Label Budget ($b$) | | | | |
|---|---|---|---|---|---|
| | 10% | 20% | 30% | 40% | 50% |
| Zero initialization | 79.96 | 87.01 | 90.32 | 91.80 | 92.52 |
| Retained Initial Weights | 80.61 | 88.22 | 91.19 | 92.75 | 93.66 |
| Random re-Initialized Weights | 80.92 | 88.35 | 91.44 | 92.77 | 93.65 |

protocol. For each pruning ratio $p$ and labeling budget $b$, we first run PruneFuse to obtain a selected labeled subset $L$. We then train the same dense target architecture (ResNet-56) on $L$ under two conditions: (i) **No fusion / No KD**: randomly initialized target trained with cross-entropy, (ii) **KD only**: no fusion; randomly initialized target trained with KD from $\theta_p^*$, (iii) **Fusion w/o KD**: target initialized via $\theta_F = \text{Fuse}(\theta, \theta_p^*)$ and trained with cross-entropy, and (iv) **Fusion + KD**: fused initialization trained with cross-entropy and KD (full PruneFuse). Importantly, the architecture, subset size, optimizer, learning-rate schedule, batch size, and total number of epochs are identical across all the conditions. In all fusion ablations, we train for 180 epochs with learning rate 0.01. Therefore, per-epoch compute is matched and using epoch as the x-axis provides a fair comparison of convergence behavior.

Figs. 5, 6, and 7 show the resulting learning curves for different pruning ratios. In the reported settings (budgets $b \in \{10\%, 30\%, 50\%\}$ and pruning ratios $p \in \{0.5, 0.6, 0.7\}$), fusion (with or without KD) consistently converges faster and reaches higher final accuracy than training from scratch on the same selected subset $L$. These results provide direct empirical evidence that fusion improves both convergence and final performance under a matched training budget.

Additionally, in Table 18 we study how the initialization of the remaining (previously untrained) weights after fusion affects performance. We compare three schemes: (i) retaining the original dense-model initialization for the non-copied weights, (ii) zero initialization, and (iii) random re-initialization. Results show that while fusion provides the primary performance gains, zeroing the remaining weights is suboptimal, and simple

randomized initializations, either retained or re-initialized, are sufficient to support effective post-fusion training.

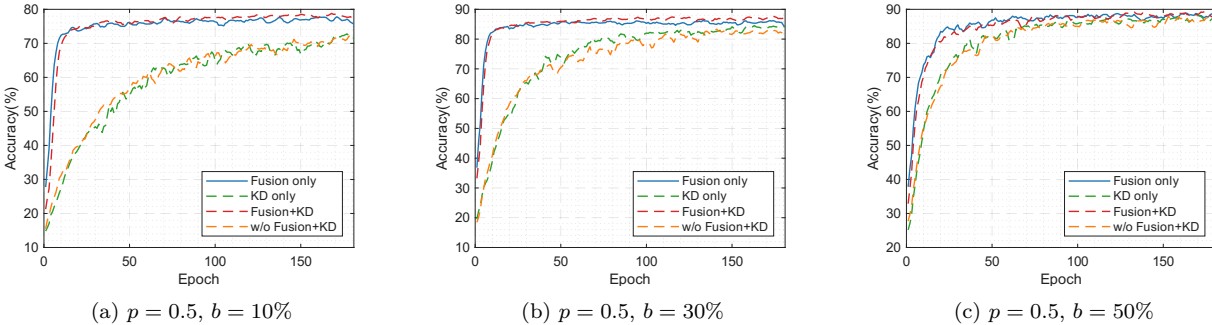

Figure 5: **Ablation Study of Fusion on PruneFuse** ($p = 0.5$). Experiments are performed on ResNet-56 architecture with CIFAR-10.

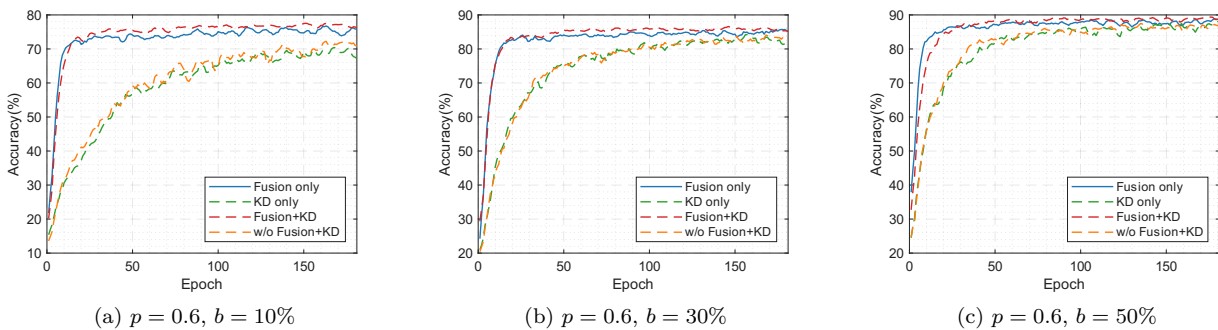

Figure 6: **Ablation Study of Fusion on PruneFuse** ($p = 0.6$). Experiments are performed on ResNet-56 architecture with CIFAR-10.

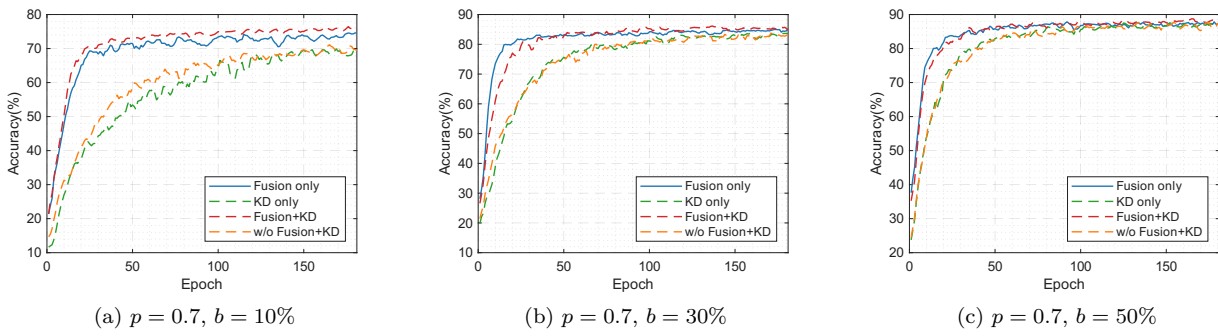

Figure 7: **Ablation Study of Fusion on PruneFuse** ($p = 0.7$). Experiments are performed on ResNet-56 architecture with CIFAR-10.

### 9.7 Ablation Study of Knowledge Distillation in PruneFuse

Table 19 demonstrates the effect of Knowledge Distillation on the PruneFuse technique relative to the baseline Active Learning (AL) method across various experimental configurations and label budgets on CIFAR-10 and CIFAR-100 datasets, using different ResNet architectures. The results indicate that PruneFuse consistently outperforms the baseline method, both with and without incorporating Knowledge Distillation (KD) from a trained pruned model. This superior performance is attributed to the innovative fusion strategy inherent to PruneFuse, where the original model is initialized using weights from a previously trained pruned model. The proposed approach gives the fused model an optimized starting point, enhancing its ability to learn more efficiently and generalize better. The impact of this strategy is evident across different label budgets and architectures, demonstrating its effectiveness and robustness.

### 9.8 Comparison with SVP

Table 21 delineates a performance comparison of PruneFuse with SVP techniques, across various labeling budgets $b$ for the efficient training of a Target Model (ResNet-56). SVP employs a ResNet-20 as its data selector, with a model size of 0.26 M. In contrast, PruneFuse uses a 50% pruned ResNet-56, reducing its data selector size to 0.21 M. Performance metrics show that as the label budget increases from 10% to 50%, the PruneFuse consistently outperforms SVP across all label budgets. Specifically on the target model, PruneFuse initiates at an accuracy of 82.68% with a 10% label budget and peaks at 93.69% accuracy at a 50% budget, whereas SVP achieves 80.76% at 10% label budget and achieves 92.95% accuracy at 50%. Notably, while the data selector of PruneFuse achieves a lower accuracy of 90.31% at $b = 50\%$ compared to SVP's 91.61%, the target model utilizing PruneFuse-selected data attains a superior accuracy of 93.69%, relative to 92.95% for the SVP-selected data. This disparity underscores the distinct operational focus of the data selectors: PruneFuse's selector is optimized for enhancing the target model's performance, rather than its own accuracy. Fig. 8(a) and (b) show that target models ResNet-14 and ResNet-20, when trained with data selectors of PruneFuse achieve significantly higher accuracy while using significantly less number of parameters compared to SVP. These results indicate that the proposed approach does not require an additional architecture for designing the data selector; it solely needs the target model (e.g. ResNet-14). In contrast, SVP necessitates both the target model (ResNet-14) and a smaller model (ResNet-8) that functions as a data selector.

Table 20 demonstrates the performance comparison of PruneFuse and SVP for small model architecture ResNet-20 on CIFAR-10. SVP achieves 91.88% performance accuracy by utilizing the data selector having 0.074 M parameters whereas PruneFuse outperforms SVP by achieving 92.29% accuracy with a data selector of 0.066 M parameters.

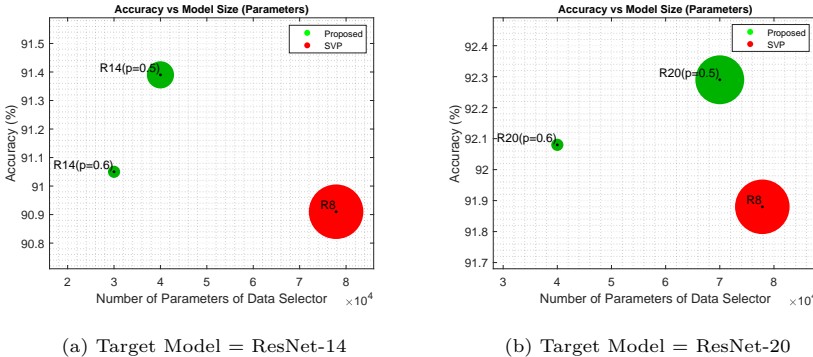

(a) Target Model = ResNet-14    (b) Target Model = ResNet-20

Figure 8: **Comparison of PruneFuse with SVP.** Scatter plot shows final accuracy on target model against the model size for different ResNet models on CIFAR-10, $b = 50\%$. (a) shows ResNet-14 (with $p = 0.5$ and $p = 0.6$) and ResNet-8 models are used as data selectors for PruneFuse and SVP, respectively. While in (b), PruneFuse utilizes ResNet20 (i.e. $p = 0.5$ and $p = 0.6$) and SVP utilizes ResNet-8 models.

Table 19: Ablation Study of Knowledge Distillation on PruneFuse presented in a, b, and c with different architectures and datasets.

| Method | Selection Metric | Label Budget ($b$) | | | | |
|---|---|---|---|---|---|---|
| | | 10% | 20% | 30% | 40% | 50% |
| Baseline AL | Least Conf | 80.53 | 87.74 | 90.85 | 92.24 | 93.00 |
| | Entropy | 80.14 | 87.63 | 90.80 | 92.51 | 92.98 |
| | Random | 78.55 | 85.26 | 88.13 | 89.81 | 91.20 |
| | Greedy k | 79.63 | 86.46 | 90.09 | 91.90 | 92.80 |
| PruneFuse $p = 0.5$ (without KD) | Least Conf | **81.08** | **88.71** | **91.24** | **92.68** | **93.46** |
| | Entropy | **80.80** | **88.08** | **90.98** | **92.74** | **93.43** |
| | Random | **80.11** | **85.78** | **88.81** | **90.20** | 91.10 |
| | Greedy k | **80.07** | **86.70** | 89.93 | 91.72 | 92.67 |
| PruneFuse $p = 0.5$ (with KD) | Least Conf | **80.92** | **88.35** | **91.44** | **92.77** | **93.65** |
| | Entropy | **81.08** | **88.74** | **91.33** | **92.78** | **93.48** |
| | Random | **80.43** | **86.28** | **88.75** | **90.36** | **91.42** |
| | Greedy k | **79.85** | **86.96** | **90.20** | 91.82 | **92.89** |

(a) CIFAR-10 using ResNet-56 architecture.

| Method | Selection Metric | Label Budget ($b$) | | | | |
|---|---|---|---|---|---|---|
| | | 10% | 20% | 30% | 40% | 50% |
| Baseline AL | Least Conf | 81.15 | 89.4 | 92.72 | 94.09 | 94.63 |
| | Entropy | 80.99 | 89.54 | 92.45 | 94.06 | 94.49 |
| | Random | 80.27 | 87.00 | 89.94 | 91.57 | 92.78 |
| | Greedy k | 80.02 | 88.33 | 91.76 | 93.39 | 94.40 |
| PruneFuse $p = 0.5$ (without KD) | Least Conf | **83.82** | **90.26** | **93.15** | **94.34** | **94.90** |
| | Entropy | **82.72** | **90.42** | **93.18** | **94.68** | **95.00** |
| | Random | **81.94** | **88.04** | **90.37** | **91.93** | 92.67 |
| | Greedy k | **81.99** | **89.04** | **92.14** | **93.40** | **94.44** |
| PruneFuse $p = 0.5$ (with KD) | Least Conf. | **83.03** | **90.30** | **93.00** | **94.41** | **94.63** |
| | Entropy | **82.64** | **89.88** | **93.08** | **94.32** | **94.90** |
| | Random | **81.52** | **87.84** | **90.14** | **91.94** | **92.81** |
| | Greedy k | **81.70** | **88.75** | **91.92** | **93.64** | 94.22 |

(b) CIFAR-10 using ResNet-164 architecture.

| Method | Selection Metric | Label Budget ($b$) | | | | |
|---|---|---|---|---|---|---|
| | | 10% | 20% | 30% | 40% | 50% |
| Baseline AL | Least Conf | 35.99 | 52.99 | 59.29 | 63.68 | 66.72 |
| | Entropy | 37.57 | 52.64 | 58.87 | 63.97 | 66.78 |
| | Random | 37.06 | 51.62 | 58.77 | 62.05 | 64.63 |
| | Greedy k | 38.28 | 52.43 | 58.96 | 63.56 | 66.30 |
| PruneFuse $p = 0.5$ (without KD) | Least Conf | **39.27** | **54.25** | **60.6** | **64.17** | **67.49** |
| | Entropy | 37.43 | **52.57** | **60.57** | **64.44** | **67.31** |
| | Random | **40.07** | **52.83** | **59.93** | **63.06** | **65.41** |
| | Greedy k | **39.25** | 52.43 | **59.94** | **63.94** | **66.56** |
| PruneFuse $p = 0.5$ (with KD) | Least Conf | **40.26** | **53.90** | **60.80** | **64.98** | **67.87** |
| | Entropy | **38.59** | **54.01** | **60.52** | **64.83** | **67.67** |
| | Random | **39.43** | **54.60** | **60.13** | **63.91** | **66.02** |
| | Greedy k | **39.83** | **54.35** | **60.40** | **64.22** | **66.89** |

(c) CIFAR-100 using ResNet-56 architecture.

## 9.9 Ablation Study on the Number of Selected Data Points ($k$)

Table 22 and 23 present ablation studies analyzing the effect of varying $k$ on the performance of PruneFuse with $T_{sync} = 0$ and $T_{sync} = 1$, respectively, on CIFAR-10 using the ResNet-56 architecture and least confidence as the selection metric. The results demonstrate that the choice of $k$ significantly impacts the quality of data selection and the final performance of the model. As $k$ increases, the selected subset quality diminishes as can be seen by comparing performance of the target network in both tables. This study highlights the importance of tuning $k$ to achieve an optimal trade-off between computational efficiency and model accuracy.

Table 20: Comparison of SVP and PruneFuse on Small Models.

| Techniques | Model | Architecture | Params (Million) | Label Budget ($b$) | | | | |
|---|---|---|---|---|---|---|---|---|
| | | | | 10% | 20% | 30% | 40% | 50% |
| SVP | Data Selector | ResNet-8 | 0.074 | 77.85 | 83.35 | 85.43 | 86.83 | 86.90 |
| | Target | ResNet-20 | 0.26 | 80.18 | 86.34 | 89.22 | 90.75 | 91.88 |
| PruneFuse | Data Selector | ResNet-20 ($p = 0.5$) | **0.066** | 76.58 | 83.41 | 85.83 | 87.07 | 88.06 |
| | Target | ResNet-20 | 0.26 | **80.25** | **87.57** | **90.20** | **91.70** | **92.29** |

Table 21: Comparison with SVP.

| Method | Model | Architecture | Params (Million) | Label Budget ($b$) | | | | |
|---|---|---|---|---|---|---|---|---|
| | | | | 10% | 20% | 30% | 40% | 50% |
| SVP | Data Selector | ResNet-20 | 0.26 | 81.07 | 86.51 | 89.77 | 91.08 | 91.61 |
| | Target | ResNet-56 | 0.85 | 80.76 | 87.31 | 90.77 | 92.59 | 92.95 |
| PruneFuse | Data Selector | ResNet-56 ($p = 0.5$) | **0.21** | 78.62 | 84.92 | 88.17 | 89.93 | 90.31 |
| | Target | ResNet-56 | 0.85 | **82.68** | **88.97** | **91.63** | **93.24** | **93.69** |

Table 22: **Ablation study of** $k$ on CIFAR-10 using ResNet-56 architecture and least confidence as a selection metric.

(a) $k = 7.5K$.

| Method | Label Budget ($b$) | | | | |
|---|---|---|---|---|---|
| | 15% | 30% | 45% | 60% | 75% |
| Baseline ($AL$) | 84.63 | 90.59 | 92.77 | 93.12 | 93.94 |
| PruneFuse ($p = 0.5$) | **85.80** | **91.13** | **93.72** | **93.84** | **94.10** |

(b) $k = 5K$.

| Method | Label Budget ($b$) | | | | |
|---|---|---|---|---|---|
| | 10% | 20% | 30% | 40% | 50% |
| Baseline ($AL$) | 80.53 | 87.74 | 90.85 | 92.24 | 93.00 |
| PruneFuse ($p = 0.5$) | **80.92** | **88.35** | **91.44** | **92.77** | **93.65** |

Table 23: **Ablation study of** $k$ on Cifar-10 using ResNet-56 with $p = 0.5$ and $T_{sync} = 1$.

| Method | Selection Size (k) | Label Budget ($b$) | | | Selection Size (k) | Label Budget ($b$) | | |
|---|---|---|---|---|---|---|---|---|
| | | 20% | 40% | 60% | | 20% | 40% | 60% |
| **Baseline** ($AL$) | 5,000 | 88.51 | 93.04 | 93.83 | 10,000 | 86.92 | 92.51 | 93.81 |
| **PruneFuse** | 5,000 | **88.52** | **93.15** | **93.90** | 10,000 | **87.49** | **93.11** | **94.04** |

## 9.10 Impact of Early Stopping on Performance

Table 24 explores the effect of utilizing an early stopping strategy alongside PruneFuse ($p = 0.5$) on CIFAR-10 with the ResNet-56 architecture. The results indicate that early stopping not only reduces training time of the fused model but also maintains comparable performance to fully trained models. This highlights the compatibility of PruneFuse with training efficiency techniques such as early stopping and showcases how the expedited convergence enabled by the fusion process further enhances its practicality, particularly in resource-constrained environments.

Table 24: **Performance Comparison** when Early Stopping strategy is utilized alongside PruneFuse ($p = 0.5$). Experiments are performed with Resnet-56 on CIFAR-10.

| Method | Epochs | Label Budget ($b$) | | | | |
|---|---|---|---|---|---|---|
| | | 10% | 20% | 30% | 40% | 50% |
| Least Conf. | 181 | 80.92±0.409 | 88.35±0.327 | 91.44±0.148 | 92.77±0.026 | 93.65±0.141 |
| | 110 | 80.51±0.375 | 87.64±0.222 | 90.79±0.052 | 92.11±0.154 | 93.00±0.005 |
| Entropy | 181 | 81.08±0.155 | 88.74±0.103 | 91.33±0.045 | 92.78±0.045 | 93.48±0.042 |
| | 110 | 80.51±0.401 | 87.46±0.416 | 90.97±0.116 | 92.2±0.108 | 92.88±0.264 |
| Random | 181 | 80.43±0.273 | 86.28±0.367 | 88.75±0.17 | 90.36±0.022 | 91.42±0.125 |
| | 110 | 79.29±0.355 | 84.99±0.156 | 87.86±0.323 | 89.99±0.090 | 90.85±0.012 |
| Greedy k. | 181 | 79.85±0.676 | 86.96±0.385 | 90.20±0.164 | 91.82±0.136 | 92.89±0.144 |
| | 110 | 79.36±0.274 | 86.36±0.455 | 89.67±0.319 | 91.19±0.302 | 91.91±0.021 |

### 9.11 Performance Comparison Across Architectures and Datasets

In Table 25, we present the performance comparison of Baseline and PruneFuse across various architectures and datasets. These results demonstrate the adaptability of PruneFuse to different network architectures, including ResNet-18, ResNet-50, and Wide-ResNet (W-28-10), as well as datasets such as CIFAR-10, CIFAR-100, and ImageNet. The experiments confirm that PruneFuse consistently improves performance over the baseline, highlighting its generalizability and robustness across diverse scenarios.

Table 25: **Performance Comparison of Baseline and PruneFuse** presented in a, b, and c with different architectures and datasets.

| Method | Label Budget ($b$) | | | | |
|---|---|---|---|---|---|
| | 10% | 20% | 30% | 40% | 50% |
| Baseline ($AL$) | 83.12 | 90.07 | 92.71 | 94.07 | 94.81 |
| PruneFuse ($p = 0.5$) | **83.29** | **90.56** | **93.17** | **94.56** | **95.08** |

(a) ResNet-18 architecture on CIFAR-10.

| Method | Label Budget ($b$) | | | | |
|---|---|---|---|---|---|
| | 10% | 20% | 30% | 40% | 50% |
| Baseline ($AL$) | 84.74 | 91.48 | 94.17 | 95.24 | 95.75 |
| PruneFuse ($p = 0.5$) | **85.65** | **92.27** | **94.65** | **95.73** | **96.24** |

(b) Wide-ResNet architecture on CIFAR-10.

| Method | Label Budget ($b$) | | | | |
|---|---|---|---|---|---|
| | 10% | 20% | 30% | 40% | 50% |
| Baseline ($AL$) | 52.97 | 64.52 | 69.30 | 71.98 | 73.56 |
| PruneFuse ($p = 0.5$) | **55.03** | **65.12** | **69.72** | **72.07** | **73.86** |

(c) ResNet-50 architecture on ImageNet-1K.

### 9.12 Performance at Lower Pruning Rates

Table 26 provides a performance comparison of Baseline and PruneFuse with a lower pruning rate of $p = 0.4$ on CIFAR-10 and CIFAR-100 using the ResNet-56 architecture. Least Confidence and Entropy were used as selection metrics for these experiments. The results show that even at a lower pruning rate, PruneFuse effectively selects high-quality data subsets, maintaining strong performance in both datasets. These findings validate the method's effectiveness across different pruning rates.

### 9.13 Comparison with Recent Coreset Selection Techniques

Table 27 compares the performance of Baseline (Coreset Selection) and PruneFuse ($p = 0.5$) using various recent selection metrics, including Forgetting Events (Toneva et al., 2019), Moderate (Xia et al., 2022), and CSS (Zheng et al., 2022) on the CIFAR-10 dataset with the ResNet-56 architecture.

To incorporate these recent score metrics, which are specifically designed for coreset-based selection, we utilized the coreset task setup. In this setup, the network is first trained on the entire dataset to identify a representative subset of data (coreset) based on the selection metric. The accuracy of the target model trained on the selected coreset is then reported. The results demonstrate that PruneFuse seamlessly integrates with these advanced selection metrics, achieving competitive or superior performance compared to the baseline while maintaining computational efficiency. This highlights the versatility of PruneFuse in adapting to and enhancing existing coreset selection techniques.

Table 26: **Performance Comparison of Baseline and PruneFuse**($p = 0.4$) on Cifar-10 and Cifar-100 using ResNet-56 architecture.

| Method | Selection Metric | Label Budget ($b$) | | | | |
|---|---|---|---|---|---|---|
| | | **10%** | **20%** | **30%** | **40%** | **50%** |
| Baseline ($AL$) | Least Confidence | 80.53 | 87.74 | 90.85 | 92.24 | 93.00 |
| | Entropy | 80.14 | 87.63 | 90.80 | 92.51 | 92.98 |
| PruneFuse ($p = 0.4$) | Least Confidence | **81.12** | **88.16** | **91.35** | **92.89** | **93.20** |
| | Entropy | **80.94** | **88.27** | **91.09** | **92.73** | **93.38** |

(a) CIFAR-10

| Method | Selection Metric | Label Budget ($b$) | | | | |
|---|---|---|---|---|---|---|
| | | **10%** | **20%** | **30%** | **40%** | **50%** |
| Baseline ($AL$) | Least Confidence | 35.99 | 52.99 | 59.29 | 63.68 | 66.72 |
| | Entropy | 37.57 | 52.64 | 58.87 | 63.97 | 66.78 |
| PruneFuse ($p = 0.4$) | Least Confidence | **38.73** | **54.35** | **60.75** | **64.80** | **67.08** |
| | Entropy | **38.35** | **54.19** | **60.79** | **65.00** | **67.47** |

(b) CIFAR-100

Table 27: **Performance Comparison of Baseline (Coreset) and PruneFuse ($p = 0.5$) for Various selection metrics** including Forgetting Events (Toneva et al., 2019), Moderate (Xia et al., 2023), and CSS (Zheng et al., 2023) on Cifar-10 dataset using ResNet-56 architecture.

| Method | Selection Metric | Data Selector's Params | Target Model's Params | Accuracy ($b = 25\%$) |
|---|---|---|---|---|
| **Baseline** | Entropy | 0.85 Million | 0.85 Million | 86.13 |
| | Least Confidence | | | 86.50 |
| | Forgetting Events | | | 86.01 |
| | Moderate | | | 86.27 |
| | CSS | | | 87.21 |
| **PruneFuse** | Entropy | 0.21 Million | 0.85 Million | **86.71** |
| | Least Confidence | | | **86.68** |
| | Forgetting Events | | | **87.84** |
| | Moderate | | | **87.63** |
| | CSS | | | **88.85** |

## 9.14 Effect of Various Pruning Strategies and Criteria

In Table 28, we evaluate the impact of different pruning techniques (e.g., static pruning, dynamic pruning) and pruning criteria (e.g., L2 norm, GroupNorm Importance, LAMP Importance (Fang et al., 2023)) on the performance of PruneFuse ($p = 0.5$) on CIFAR-10 using the ResNet-56 architecture.

Static pruning involves pruning the entire network at once at the start of training, whereas dynamic pruning incrementally prunes the network in multiple steps during training. In our implementation of dynamic pruning, the network is pruned in five steps over the course of 20 epochs.

The results demonstrate that PruneFuse is highly adaptable to various pruning strategies, consistently maintaining strong performance in data selection tasks. This flexibility underscores the robustness of the framework across different pruning approaches and criteria.

## 9.15 Runtime Comparison of Data Selector Networks and Detailed Breakdown of the Training Runtime for each Component of PruneFuse

Table 29 compares the training runtimes of the data selector network (pruned network for PruneFuse and dense network for the baseline) across various network architectures. The reported times correspond to the training phase of the data selector network prior to the final selection of the subset (at $b = 50\%$, label budget). Note that the variation in runtimes across different datasets is due to the experiments being conducted on different servers, each equipped with specific GPUs (e.g., 2080Ti, 3090, or A100). The results show that

Table 28: **Effect of Pruning techniques and Pruning criteria** on PruneFuse ($p = 0.5$) on CIFAR-10 dataset with ResNet-56.

| Method | Pruning Technique | Pruning Criteria | Label Budget ($b$) | | | | |
|---|---|---|---|---|---|---|---|
| | | | 10% | 20% | 30% | 40% | 50% |
| Baseline (AL) | - | - | 80.53 | 87.74 | 90.85 | 92.24 | 93.00 |
| PruneFuse ($T_{sync} = 0$) | Dynamic Pruning | Magnitude Imp. | 79.73 | 87.16 | **91.08** | 92.29 | 93.19 |
| | | GroupNorm Imp. | 80.10 | 88.25 | 91.01 | 92.25 | 93.74 |
| | | LAMP Imp. | **81.51** | 87.45 | 90.64 | **92.41** | 93.25 |
| PruneFuse ($T_{sync} = 0$) | Static Pruning | Magnitude Imp. | 80.92 | 88.35 | 91.44 | 92.77 | 93.65 |
| | | GroupNorm Imp. | 80.84 | 88.20 | 91.19 | 93.01 | 93.03 |
| | | LAMP Imp. | 81.10 | 88.37 | 91.32 | 93.02 | 93.08 |
| PruneFuse ($T_{sync} = 1$) | Static Pruning | Magnitude Imp. | 81.23 | 88.52 | 91.76 | 93.15 | 93.78 |
| | | GroupNorm Imp. | 81.09 | 88.77 | 91.77 | 93.19 | 93.68 |
| | | LAMP Imp. | 81.86 | 88.51 | 92.10 | 93.02 | 93.63 |

PruneFuse significantly reduces training time due to the efficiency of the pruned network as compared to baseline, making it well suited for resource-constrained environments.

Table 30 provides a detailed breakdown of the training run time for each component of PruneFuse, including the data selector training time, the selection time, and the target network training time. These measurements offer a comprehensive view of the computational requirements of PruneFuse, demonstrating its efficiency compared to the baseline methods. The breakdown highlights that the pruned network and the fusion process contribute to significant computational savings without compromising performance.

Table 29: **Training Runtime** of data selector network i.e. pruned network in the case of PruneFuse and dense network for baseline, for various network architectures. The reported time is the training time when the network is trained before selecting final subset of the data ($b = 50\%$).

| Datasets | Data Selectors (Selection Models) | Training Runime (Minutes) |
|---|---|---|
| CIFAR-10 | ResNet-56 (Baseline) | 127.67 |
| | ResNet-56 (PruneFuse ($p = 0.5$)) | **72.55** |
| | ResNet-56 (PruneFuse ($p = 0.8$)) | **67.23** |
| | ResNet-18 (Baseline) | 85.68 |
| | ResNet-18 (PruneFuse ($p = 0.5$)) | **61.15** |
| | Wide ResNet (Baseline) | 122.43 |
| | Wide ResNet (PruneFuse ($p = 0.5$)) | **75.48** |
| CIFAR-100 | ResNet-164 (Baseline) | 129.23 |
| | ResNet-164 (PruneFuse ($p = 0.5$)) | **83.52** |
| | ResNet-164 (PruneFuse ($p = 0.8$)) | **78.55** |
| | ResNet-110 (Baseline) | 95.80 |
| | ResNet-110 (PruneFuse ($p = 0.5$)) | **80.42** |
| | ResNet-110 (PruneFuse ($p = 0.8$)) | **69.50** |
| TinyImagenet-200 | ResNet-50 (Baseline) | 248.48 |
| | ResNet-50 (PruneFuse ($p = 0.5$)) | **147.47** |
| | ResNet-50 (PruneFuse ($p = 0.8$)) | **94.42** |
| ImageNet-1K | Resnet-50 (Baseline) | 2081.3 |
| | ResNet-50 (PruneFuse ($p = 0.5$)) | **951.17** |

Table 30: **Detailed Training time of Baseline and PruneFuse($p = 0.5$) for TinyImageNet-200 for Resnet-50** using Least Confidence as selection metric.

| Datasets | Label Budget ($b$) | Data Selectors (Training Time) (Minutes) | Data Selection Time (Minutes) | Target Model (Training Time) (Minutes) |
|---|---|---|---|---|
| **Baseline (AL)** | 10% | 48.80 | 4.43 | 48.80 |
| | 20% | 99.23 | 3.50 | 99.23 |
| | 30% | 145.32 | 3.15 | 145.32 |
| | 40% | 195.38 | 2.72 | 195.38 |
| | 50% | 248.48 | 2.38 | 248.48 |
| **PruneFuse** | 10% | **32.17** | **1.57** | 49.50 |
| | 20% | **61.70** | **1.67** | 99.99 |
| | 30% | **88.53** | **1.52** | 146.25 |
| | 40% | **117.10** | **1.37** | 196.28 |
| | 50% | **147.47** | **1.18** | 249.58 |

Table 31: **Results with ResNet20 and ResNet56 on CIFAR-10 (Least Confidence & Entropy) using ALSE Jung et al. (2023).**

| Model / Selection Metric | Method with ALSE | 10% | 20% | 30% | 40% | 50% |
|---|---|---|---|---|---|---|
| ResNet20 / Least Conf. | Baseline | 80.24 | 86.28 | 89.07 | 90.27 | 91.09 |
| | PruneFuse ($p = 0.4$) | **81.05** | **86.82** | **90.02** | **90.89** | **91.54** |
| | PruneFuse ($p = 0.5$) | **80.72** | **86.65** | **89.79** | **90.79** | **91.51** |
| | PruneFuse ($p = 0.6$) | **80.53** | **86.68** | **89.56** | **90.81** | **91.32** |
| ResNet56 / Least Conf. | Baseline | 80.73 | 88.13 | 90.99 | 92.58 | 93.13 |
| | PruneFuse ($p = 0.4$) | **80.86** | **88.28** | **91.36** | **93.15** | **93.44** |
| | PruneFuse ($p = 0.5$) | **80.80** | **88.17** | **91.43** | **93.02** | **93.19** |
| | PruneFuse ($p = 0.6$) | **80.76** | **87.80** | **91.29** | **92.73** | **93.20** |
| ResNet20 / Entropy | Baseline | 80.12 | 86.01 | 88.96 | 90.68 | 91.21 |
| | PruneFuse ($p = 0.4$) | **80.64** | **86.99** | **89.65** | **91.27** | **91.46** |
| | PruneFuse ($p = 0.5$) | **80.39** | **86.59** | **89.49** | **90.97** | **91.42** |
| | PruneFuse ($p = 0.6$) | **80.29** | **86.24** | **88.94** | **90.75** | **91.24** |
| ResNet56 / Entropy | Baseline | 80.59 | 88.11 | 90.88 | 92.52 | 93.06 |
| | PruneFuse ($p = 0.4$) | **81.05** | **88.49** | **91.38** | **92.95** | **93.44** |
| | PruneFuse ($p = 0.5$) | **80.97** | **88.37** | **91.31** | **92.87** | **93.32** |
| | PruneFuse ($p = 0.6$) | **80.77** | **88.09** | **91.08** | **92.71** | **93.20** |

