# OpenReview forum: "PruneFuse: Efficient Data Selection via Weight Pruning and Network Fusion"
_TMLR — Accepted by TMLR_

### Review · Reviewer_9M11 · 2025-12-13

**Summary Of Contributions:**

This paper proposes PruneFuse, a framework for efficient data selection in active learning settings by leveraging structured pruning and weight-aligned network fusion. The core idea is to use a pruned network at initialization as a lightweight proxy for data selection, thereby significantly reducing the computational cost of active learning, and then to fuse the trained pruned model back into the original dense network instead of discarding it. This fusion provides a warm-start initialization for the final model and can be optionally combined with knowledge distillation and periodic synchronization to further improve convergence and performance.

The paper demonstrates the effectiveness of PruneFuse across a wide range of vision and text benchmarks, including CIFAR, Tiny-ImageNet, ImageNet-1K, and Amazon Reviews, as well as across multiple architectures such as ResNets, Vision Transformers, and VDCNN. Empirically, the method consistently achieves comparable or improved accuracy relative to standard active learning baselines while reducing the computational cost of the data selection stage by a large margin.

Key strengths of the work include its practical relevance, broad experimental validation, and a clear systems-level motivation addressing the scalability limitations of conventional active learning pipelines. Key weaknesses include limited causal isolation of the individual components (e.g., fusion, synchronization, and distillation), and a theoretical analysis that relies on strong assumptions which are not fully justified or empirically validated.

**Additional Comments:**

This is a well-executed and practically motivated paper that addresses a real bottleneck in active learning. With clearer positioning relative to prior work, stronger causal analysis of its core components, and a more cautious treatment of the theoretical section, the work has the potential to become a solid contribution suitable for TMLR.

**Audience:**

Yes

**Audience Explanation:**

Yes. The problem of computationally efficient data selection is of broad interest to the machine learning community, particularly for researchers and practitioners working on active learning, data-efficient training, and scalable learning systems. The idea of reusing pruned models not only as proxies for data selection but also as informative initializations for the final model is likely to be of interest to TMLR’s audience, especially given the growing emphasis on efficiency and sustainability in large-scale machine learning.

Moreover, the framework is general and compatible with existing acquisition functions and model architectures, which further increases its relevance beyond a narrow application domain.

**Broader Impact Concerns:**

I do not identify any significant ethical concerns specific to this work. The proposed method focuses on improving the computational efficiency of data selection and model training, which may have positive environmental and resource-related implications. The approach does not introduce new risks related to data privacy, misuse, or social harm beyond those already associated with standard supervised learning and active learning pipelines. A separate Broader Impact Statement does not appear to be necessary.

**Claims And Evidence:**

Yes

**Claims Explanation:**

Overall, the empirical claims of the paper are supported by extensive experimental evidence. The authors evaluate PruneFuse on multiple datasets, model architectures, pruning ratios, and labeling budgets, and consistently report both accuracy and computational cost metrics. The experimental results clearly demonstrate that the proposed approach can substantially reduce the cost of data selection while maintaining or improving downstream performance relative to standard active learning baselines.

That said, while the evidence is sufficient to support the main empirical claims, some of the stronger interpretive claims—particularly regarding the specific role of model fusion in improving optimization trajectories and exploration of the loss landscape—are not fully substantiated by targeted ablation or diagnostic analyses. These claims would benefit from more controlled experiments or from being stated in a more measured manner. Nonetheless, the core performance and efficiency claims are convincingly supported.

**Requested Changes:**

Below I list the requested changes, indicating which are critical for acceptance versus those that would strengthen the work.

Critical (required for acceptance):
	1.	Clarify the conceptual novelty relative to prior proxy-based active learning methods.
The relationship between PruneFuse and prior approaches such as Selection via Proxy (SVP) should be articulated more sharply. In particular, the authors should clearly state what is fundamentally enabled by weight-aligned fusion that cannot be achieved by proxy-based selection followed by standard training or distillation.
	2.	Provide stronger causal evidence for the role of model fusion.
While empirical results show that fusion improves performance, the current experiments do not sufficiently isolate fusion from other factors such as training budget, distillation, or initialization effects. Additional ablations or more controlled comparisons (e.g., fusion vs. no-fusion under matched training cost) are necessary to support claims about the benefits of fusion.
	3.	Revisit and clarify the theoretical analysis in Section 5.
The error bound relies on strong assumptions—particularly the contraction assumption after synchronization—that are not justified by the method’s mechanics. The authors should either empirically validate these assumptions, substantially weaken the claims of the theory, or clearly position the analysis as an intuitive or qualitative justification rather than a formal guarantee.
	4.	Address fairness of baselines in computational cost comparisons.
It should be made explicit whether the active learning baselines are FLOPs-matched or represent canonical implementations. Providing a FLOPs-matched baseline or clarifying the comparison protocol would reduce potential concerns about bias in efficiency comparisons.

Non-critical (would strengthen the paper):
	1.	Improve the clarity of the method description by separating core components from optional extensions (e.g., synchronization and knowledge distillation).
	2.	Move detailed fusion equations and layer-wise implementation specifics to the appendix, focusing the main text on intuition and design rationale.
	3.	Slightly tone down speculative language regarding loss landscape exploration unless supported by additional analysis.
	4.	Add a concise summary figure or table highlighting which components are essential versus optional in the PruneFuse pipeline.

---

> ### Author Response · Authors · 2026-01-16
> **Author Response (1/2)**
>
> We thank the reviewer for the careful and constructive feedback, and for recognizing the practical relevance and empirical breadth of PruneFuse. We revised the manuscript to address the concerns regarding conceptual novelty, fusion analysis, and theoretical assumptions, along with the remaining clarity and presentation suggestions. All references correspond to the revised manuscript and changes are highlighted in **blue**.
>
> >_1.	Clarify the conceptual novelty relative to prior proxy-based active learning methods. The relationship between PruneFuse and prior approaches such as Selection via Proxy (SVP) should be articulated more sharply. In particular, the authors should clearly state what is fundamentally enabled by weight-aligned fusion that cannot be achieved by proxy-based selection followed by standard training or distillation._
>
> **Response:** We revised the manuscript to more sharply distinguish PruneFuse from prior proxy-based selection (e.g., SVP). We now emphasize in _Section 2_ that the key capability enabled by weight-aligned fusion is direct parameter reuse: our selector is produced by structured pruning of the target architecture, yielding a channel-aligned subnetwork with a one-to-one parameter correspondence to the dense model. This makes fusion a well-defined one-shot warm-start that transfers the selector’s learned features into the target, reusing selection-time training compute and accelerating convergence.
> In contrast, SVP-style proxies are typically architecturally different, so there is no direct parameter mapping that would allow such fusion; knowledge can only be transferred indirectly (e.g., via distillation).
> We also clarified that fusion is fundamentally different from KD and added fusion/KD ablations (_Figs. 5–7_) showing that KD alone underperforms fusion, while fusion provides the primary acceleration and KD is an optional refinement when combined with fusion.
>
> >_2.	Provide stronger causal evidence for the role of model fusion. While empirical results show that fusion improves performance, the current experiments do not sufficiently isolate fusion from other factors such as training budget, distillation, or initialization effects. Additional ablations or more controlled comparisons (e.g., fusion vs. no-fusion under matched training cost) are necessary to support claims about the benefits of fusion._
>
> **Response:** We agree that the benefit of fusion should be isolated from other components as suggested. In the revised manuscript, we updated _Fig. 4_ (and Supplementary _Figs. 5–7_) to explicitly disentangle these components under a controlled comparison. For each pruning ratio, we train the same dense target model on labeled set using an identical training schedule, and compare four variants: (i) No fusion / No KD, (ii) KD only (no fusion; dense initialized randomly and trained with $θ_p^*$ as teacher), (iii) Fusion w/o KD ($θ_F$ trained with cross-entropy only), and (iv) Fusion + KD (full PruneFuse). Fusion itself is a one-shot parameter copy and incurs negligible overhead relative to dense model training.
>
> Across different pruning ratios, these ablations show that fusion provides the primary warm-start and convergence benefit compared to training from scratch on the same selected subset, while KD yields an additional but complementary improvement in various settings. We also retain _Table 19_ (KD vs no-KD across datasets/architectures) and _Table 3 / Fig. 3_ (Tsync sweep) to separately characterize the contribution of KD and synchronization.
>
> Together, these experiments provide causal evidence that the gains attributed to PruneFuse arise primarily from weight-aligned fusion, rather than from increased training budget, distillation alone, or other confounding factors.

---

> ### Author Response · Authors · 2026-01-16
> **Author Response (2/2)**
>
> >_3.	Revisit and clarify the theoretical analysis in _Section 5_. The error bound relies on strong assumptions—particularly the contraction assumption after synchronization—that are not justified by the method’s mechanics. The authors should either empirically validate these assumptions, substantially weaken the claims of the theory, or clearly position the analysis as an intuitive or qualitative justification rather than a formal guarantee._
>
> **Response:** We agree that the prior analysis relied on a contraction-after-synchronization assumption that is difficult to justify in an iterative active learning setting.  In the revised manuscript, we rewrote Section 5 entirely and explicitly positioned the analysis as an intuitive error decomposition. Further, we replace the contraction assumption with a softer proxy–target proximity premise i.e., after synchronization, the acquisition proxy $\theta_p^t$ is derived from the current dense target $\theta^t$, and we assume $|\theta_p^t-\theta^t|≤ρ_t$. This yields a bound that decomposes the representativeness error into an intrinsic selection term $\delta$ and an additional proxy–target mismatch term $ρ_t$.
>
> To support the proximity premise empirically, we evaluate proxy–target alignment by comparing the synchronized proxy against a fresh proxy trained from scratch at the same cycle, and consistently observe improved proxy–target agreement (Supplementary _Section 8.2.1_). Overall, we feel the revised analysis clarifies how synchronization affects proxy-based selection by controlling proxy–target mismatch.
>
>
> >_4.	Address fairness of baselines in computational cost comparisons. It should be made explicit whether the active learning baselines are FLOPs-matched or represent canonical implementations. Providing a FLOPs-matched baseline or clarifying the comparison protocol would reduce potential concerns about bias in efficiency comparisons._
>
> **Response:** We clarified the baseline comparison protocol in _Section 6.2_ of the revised manuscript to address fairness. All methods are evaluated under a canonical active learning setup with the same target architecture (e.g. ResNet-56), labeled budgets, and training schedule, using the selector constructions and acquisition functions defined by each method. Importantly, the comparisons are not biased in favor of PruneFuse: for example, SVP uses a ResNet-20 proxy with slightly higher capacity than our pruned selector, and BALD/ALSE use the full target model as the selector. Even under these baseline-favorable settings, PruneFuse achieves higher accuracy while requiring less selection-side computation, demonstrating fair and robust efficiency gains.
>
> >_Non-critical: 1. Improve the clarity of the method description by separating core components from optional extensions (e.g., synchronization and knowledge distillation). 2. Move detailed fusion equations and layer-wise implementation specifics to the appendix, focusing the main text on intuition and design rationale. 3. Slightly tone down speculative language regarding loss landscape exploration unless supported by additional analysis. 4. Add a concise summary figure or table highlighting which components are essential versus optional in the PruneFuse pipeline._
>
> **Response:** We thank the reviewer for these constructive suggestions.
> We revised the manuscript (_Section 4.4, 4.5, and 4.6_) to streamline the presentation of PruneFuse by clearly distinguishing its essential components from optional enhancements and focusing the main text on intuition and design principles. Technical fusion details are moved to the Supplementary Materials (_Section 8.5_), speculative discussion in _Section 4.4_ is toned down and supported with additional empirical evidence, and a concise summary (_Table 1_) is added to clarify the essential components.
>
> We believe these revisions address the reviewer’s concerns and strengthen the clarity and rigor of the paper. We thank the reviewer again for the thoughtful feedback.

---

### Review · Reviewer_Kwk8 · 2025-12-23

**Summary Of Contributions:**

Authors propose a special scheme of active learning that can be specified roughly as:
1. Select and initialise architecture suitable for the downstream task
2. Perform structural pruning to select a small subnetwork
3. Train subnetwork with several rounds of active learning
4. Fuse subnetwork to the original unpruned network
5. Finetune the resulting network on the dataset selected in 3.
6. If the computation budget is not exhausted go to 2.

Authors evaluate their scheme on several datasets and compare accuracy and computation load with various active learning baselines.

**Audience:**

Yes

**Audience Explanation:**

Active learning is a classical widely applicable direction in machine learning and statistics. Given that, present contributions can be of interest for many researchers.

**Claims And Evidence:**

Yes

**Claims Explanation:**

To an extent, authors support their claims by experimental evidence. To me, debatable claims are:
1. The importance of distillation.
2. The benefits of fuse steps.
3. The usefulness of theoretical analysis.

**Requested Changes:**

**Unclear parts**

Several parts of the main text are unclear:

1. Problem setting given in equations (1) and (2)

   Equation (1) formulate a problem as finding a subset of data sufficient for training an accurate model. The result of optimization problem (1) is a subset of data. Next, authors "modify" problem (1) and formulate it as problem (2). I find two problematic parts in this framing.

   First, in equation (2) authors use $E_{(x, y)\in s_{p}}\left[l(x, y; \theta, \theta_p)\right]$ and call this expression "expected loss on subset $s_p$ (selected using $\theta_{p}$)". If $s_p$ is selected based on $\theta_p$ then the resulting expectation is not a function of $s_p$. Given that, it is not clear how to understand the minimization problem $\arg\min_{s_p}$.

   Second, from the rest of the article it becomes clear that authors do not modify problem (1). In place of that they suggest a specific way to approximately solve problem (1). Their solution is to train progressively a set of reduced models obtained with pruning from a large base model (see Algorithm 1).

   I suggest authors clarify the relation between problems (2) and (1) and their setup overall.

2. Algorithm 1

   I find several unclear parts in Algorithm 1:

   a. Among inputs to the algorithm authors list: (i) prune model $\theta_p$, (ii) fuse model $\theta_{F}$, (iii) scored data samples $D_{j}$. All these objects are later computed in the body of the algorithm. Besides that, they all depend on the training dynamics, evaluation of trained models, initialisation of the model, etc. What do authors mean when they claim the user should provide all that as an input to the algorithm?

   b. Algorithm 1 does not reference distillation that should be added to the line with fine tuning of the base model.

   c. On page 7 authors write "Meanwhile, the rest of the untrained weights in $\theta$ still have their initial values, offering an element of randomness. This duality fosters a richer exploration of the loss landscape during subsequent training." Does it mean that we still need to keep original parameters from the initialisation even when we already perform finetuning of the based model for the first time? If randomness is important, why not generate random weights during the fuse step?

3. Figure 2: Evolution of training trajectories.

   As I understand Figure 2 provides a schematic of convergence that does not correspond to any real optimisation process. In light of the relative unimportance of fuse step (see below) it is unclear how authors can back up this illustration. Can the authors comment why they believe this illustration is important and how it would be possible to test it experimentally?

**Importance of fuse step**

Fuse step is a primal innovation of the research. The ablation of the fuse step is given in Figure 3, Table 2 and Section 8.5.

Results from Figure 3 indicate that the training without fuse step ($T_{\text{syn}} = 0$) requires less FLOPs for all pruning ratios and leads to the same or better accuracy for pruning ratios $0.5, 0.7, 0.8$. Only for $p=0.6$ the performance of training without fuse steps is noticeably worse.

Similarly, from Table 2 one can conclude that the fuse step is not important since best results are observed for $T_{\text{syn}} = 0$.

In Section 8.5. authors perform experiments that compare training dynamics with and without fusion. The precise setup of the experiment is not available in Section 8.5. Can the authors explain what kind of version of Algorithm 1 they use to obtain data presented on Figure 5? It is also not clear whether one can use epoch as $x$ axis, because methods with and without fusion will require different numbers of resources for training per epoch. Can the author take this into account in their analysis?

**Importance of distillation**

Ablation on distillation is discussed in Section 8.6. The results presented in Table 15 indicate that knowledge distillation is not important: sometimes it results in better performance, sometimes in worse performance. Do authors agree with this conclusion? If the answer is positive, can the authors clarify why description of knowledge distillation in the main text (Section 4.5) is needed?

**Theoretical results**

Theoretical results are covered in Section 5. The main result is a straightforward consequence of three assumptions: assumption 5.1, assumption 5.2, assumption 5.3.

1. The first assumption 5.1. requires the neural network to be Lipschitz with respect to model parameters, which is an unusual but in principle enforceable and testable assumption. Did the authors try to estimate Lipschitz constant for neural networks they trained? What is a typical Lipschitz constant (wrt parameters) for ResNet-110?

2. Assumption 5.2. is quite strong and completely unjustified. In plain English this assumption states that: (i) a reduced subset of a full dataset exists, (ii) scores of pruned networks are enough to find that subset. Can the authors explain why they make such an assumption? Is it possible to test this assumption experimentally?

3. Similarly, Assumption 5.3. is way too strong. It implies that synchronisation steps lead to linear convergence to the parameters of the full model. In principle such an assumption could be tested, with a caveat that the same function can be represented by neural networks with parameters that are far apart in the $L_2$ norm. For example, for neural network with two layers
   $$
   W^{2}\text{ReLU}\left(W^{1}x + b^{1}\right) + b^{2} = sW^{2}\text{ReLU}\left(\frac{1}{s}W^{1}x + \frac{1}{s}b^{1}\right) + b^{2},
   $$
   for arbitrary positive $s$. The distance between parameters of these two networks increases with the increase of $s$ (for sufficiently large $s$) if weights $W^{2}$ are nonzero. Besides that, the statement should take into account somehow that the loss landscape can have a large number of distinct global minima.

Can the authors comment on the naturalness of assumption and on the purpose of given theoretical results? Why are these results useful and what they explain?

---

My main takeaways after reading the paper are:

1. Fuse step is not important.

2. Distillation is not important.

3. Samples selected with active learning using small models transfer reasonably well to larger structurally similar models.

I find point 3. to be the main contribution of the present research. I kindly ask authors to discuss this (agree/disagree, provide arguments) in the rebuttal.

---

> ### Author Response · Authors · 2026-01-17
> **Author Response (1/3)**
>
> We thank the reviewer for the careful and detailed feedback. The comments were very helpful in improving the clarity of the problem formulation, algorithm description, and scope of the theoretical analysis. We have revised the manuscript accordingly, with all the changes highlighted in **blue**.
>
> >_1.	Problem setting given in equations (1) and (2) ..._
>
> **Response:**
>
> We agree that the original framing of _Eq. (2)_ is unclear and we appreciate the reviewer pointing this out. In the revised manuscript, we restated the problem setup to clearly distinguish the ideal objective from the algorithmic approximation.
>
> _Eq. (1)_ now cleanly defines the ideal subset selection objective. PruneFuse does not modify this objective. Instead, it proposes a proxy-constrained strategy to approximately solve _Eq. (1)_. The revised _Eq. (2)_ explicitly formulates this as a constrained variant of _Eq. (1)_, where the optimization variable remains the selected subset and the proxy appears only through the acquisition constraint.
>
> We also clarify that PruneFuse is an algorithmic procedure (_Algorithm 1_) based on progressively trained pruned models, rather than a reformulation of the learning objective itself. We believe this revised presentation resolves the concerns raised by the reviewer.
>
> >_2.	Algorithm 1: I find several unclear parts in Algorithm 1:
> a. Among inputs to the algorithm ...
> b. Algorithm 1 does not reference distillation ...
> c. On page 7 authors write "Meanwhile, the rest of the untrained weights .." Does it mean that we still need to keep original parameters from the initialisation ..._
>
> **Response:**
>
> **a) and b)** We updated _Algorithm 1_ to correctly distinguish user-provided inputs from intermediate variables computed within the procedure, and revised the fine-tuning step to explicitly include KD (as an optional component), consistent with the updated method description.
>
> **c)** We have clarified this point in the revised manuscript. Keeping the original initialization for the non-copied weights is not required and these weights can be re-initialized at the fusion step, and we treat this choice as an implementation option rather than a requirement for “exploration.” We revised _Section 4.4_ to avoid implying that the effect is uniquely tied to preserving the initial parameters. Also note that our default PruneFuse implementation uses random re-initialization for the non-copied weights at the fusion step.
> Furthermore, we have added an explicit ablation study in the Supplementary (_Section 8.6, Table 18_) comparing three strategies for the remaining (non-copied) weights after fusion: retaining the original initialization, zero initialization, and random re-initialization. The results show that zero initialization performs markedly worse, while retaining the original initialization and random re-initialization achieve comparable and substantially better performance.
>
> >_3.	Figure 2: Evolution of training trajectories.
> As I understand Fig. 2 provides a schematic of convergence ..._
>
> **Response:**
>
> We agree that _Fig. 2_ is schematic and a conceptual illustration of the optimization trajectory. Its purpose is to provide intuition for two aspects of PruneFuse: (i) pruning at initialization restricts optimization to a structured subnetwork parameterization, and (ii) weight-aligned fusion embeds a trained subnetwork solution into the dense model to form a warm-start initialization, after which training proceeds in the full parameter space.
>
> To avoid ambiguity, we revised the caption and surrounding text to explicitly state that _Fig. 2_ is a conceptual illustration rather than an empirical trajectory. We also clarify how this intuition can be tested experimentally i.e.  in the Supplementary Materials (_Sec. 8.6_), we compare learning curves with and without fusion under matched conditions (same subset, architecture, optimizer, and training schedule), and observe consistently faster convergence and improved final accuracy with fusion. These ablations provide direct empirical evidence supporting the qualitative behavior summarized by _Fig. 2_.

---

> ### Author Response · Authors · 2026-01-17
> **Author Response (2/3)**
>
> >_4.	Fuse step is a primal innovation of the research. The ablation of the fuse step is given in Figure 3, Table 2 (now Table 3) and Section 8.6. ..._
>
> **Response:**
>
> To clarify the reviewer’s interpretation, we distinguish between
>
> (A) intermediate **Synchronization**, controlled by ($T_{\text{sync}}$) and
>
> (B) the final **Fusion** step used to initialize the dense target model after subset selection.
>
> In PruneFuse, the final fusion step is always applied: we initialize the final dense model ($\theta_F$) by fusing the trained pruned selector ($\theta_p^*$) into the dense model (_Sec. 4.4_). The parameter ($T_{\text{sync}}\in{0,1,2}$) controls only whether we additionally perform intermediate “fuse $\rightarrow$ re-prune $\rightarrow$ retrain” updates during the active learning rounds (_Sec. 4.6_).  With $T_{\text{sync}}=0$ representing no intermediate synchronization; we prune only once at initialization and fuse only at the end of a particular budget selection. Accordingly, _Fig. 3_ and _Table 3_ evaluate synchronization schedules (including the case ($T_{\text{sync}}=0$), i.e., no intermediate synchronization), and not the “fusion vs. no fusion”.
>
> The true “with fusion vs. without fusion” comparison is reported separately in our fusion ablations (main text _Fig. 4_ and Supplementary _Sec. 8.6_). In these experiments, we first run PruneFuse to select a labeled subset ($L$) and train the same dense ResNet-56 on ($L$) under an identical training configuration while varying only the initialization and/or KD: (i) no fusion/no KD (random initialization), (ii) KD only, (iii) fusion without KD ($\theta_F=\mathrm{Fuse}(\theta,\theta_p^*$)), and (iv) fusion+KD. Optimizer, learning-rate schedule, batch size, and total epochs are identical across variants (180 epochs with $lr=0.01$ in _Section 8.6_), so per-epoch training compute is matched; using epoch as the x-axis is therefore appropriate for comparing convergence. Under this controlled setup, fusion (with or without KD) converges faster and reaches higher final accuracy than training from scratch on the same selected subset.
>
> To address the reviewer’s request, we have updated Supplementary _Section 8.6_ to state this protocol explicitly and cross-referenced it from the main text to clear out the confusion.
>
> >_5.	Ablation on distillation is discussed in Section 8.6 (now 8.7). The results presented in Table 15 indicate that ..._
>
> We agree with the reviewer’s interpretation that knowledge distillation (KD) is not a core component of PruneFuse and its effect can be marginal or neutral in some settings. In the revised manuscript, we updated the discussion in _Section 4.5_ to clearly present KD as an optional refinement, rather than a fundamental component, and distinguish it explicitly from the core fusion mechanism.
> Consistent with prior observations on KD, our ablations indicate that KD tends to be more beneficial on larger or more complex datasets (e.g., CIFAR-100), where the teacher can convey richer information, while its impact is limited on smaller datasets. Accordingly, we revised the main text to emphasize that PruneFuse does not rely on KD to achieve its main performance gains, but that KD can be incorporated when beneficial, in alignment with the empirical findings in _Section 8.7_.

---

> ### Author Response · Authors · 2026-01-17
> **Author Response (3/3)**
>
> >_6.	 Theoretical results are covered in Section 5. The main result is a straightforward consequence of three assumptions: a) assumption 5.1, b) assumption 5.2, c) assumption 5.3 ...
> Can the authors comment on the naturalness of assumption and on the purpose of given theoretical results? Why are these results useful and what they explain?_
>
> **Response:**
>
> We agree that the earlier theoretical formulation relied on somewhat strong assumptions. In the revised manuscript, we have rewritten _Section 5_ as an error-decomposition and revised the assumptions as below:
>
> **a)** Assumption 1 (Lipschitz continuity of the loss w.r.t. parameters) is a regularity condition often used in optimization and FL analyses (e.g., FedAPA[1]) to relate parameter mismatch to loss mismatch. We assume it only locally on the region of parameter space explored during training and do not estimate the constant in practice.
>
> **b)** Assumption 2 (a δ-coreset / representativeness premise under the selector) is standard in coreset-style subset selection / active learning [2]. It formalizes the common redundancy assumption underlying AL that real datasets are typically redundant and a budget-k subset selected by an acquisition rule should approximate the full pool under the selector model, up to an intrinsic error δ (a selection floor that also exists in classical AL). We adopt this premise because PruneFuse is orthogonal to the specific acquisition function (entropy, least-confidence, k-centers, etc.). The analysis treats the acquisition rule as a black box and summarizes its inherent approximation quality through δ. While δ could in principle be estimated, its value depends strongly on the acquisition rule and data distribution, making it difficult to interpret in a general or comparable way.
>
> Ultimately, the relevant consequence of a small δ is that a selected subset enables target models to perform comparable to those trained on the full dataset. Our experiments directly validate this outcome consistently across acquisition rules, datasets, and architectures.
>
> **c)** We replaced the previous contraction-style _Assumption 3_ with a soft proxy–target proximity condition. It simply captures the degree of alignment between the proxy used for selection and the target being evaluated. We support this premise empirically by showing that the synchronized proxy remains closer to the target than a fresh proxy, i.e.,$\Delta_t^{\mathrm{sync}}\le \Delta_t^{\mathrm{fresh}}$ (see Supplementary _Section 8.2.1, Table 8_).
>
> Overall, the revised theory is intended as an interpretive error decomposition highlighting the two factors that govern proxy-based selection quality (i) acquisition/coreset error δ and (ii) proxy–target mismatch, thereby motivating synchronization as a mechanism to control the latter.
>
> >_7. My main takeaways after reading the paper are:
> _1.	Fuse step is not important._
> 2.	Distillation is not important.
> 3.	Samples selected with active learning using small models transfer reasonably well to larger structurally similar models.
> I find point 3. to be the main contribution of the present research. I kindly ask authors to discuss this (agree/disagree, provide arguments) in the rebuttal._
>
> **Response:**
>
> We appreciate the reviewer’s thoughtful summary and address the three points briefly, referring to our earlier clarifications above.
>
> **1.**	_Distillation is not important_ (partially agree)
>
> We agree that knowledge distillation is not a core component of PruneFuse and is best viewed as an optional refinement, as reflected in our ablations and revised discussion.
>
> **2.**	_Fuse step is not important_ (disagree)
>
> We respectfully clarify that fusion is an essential component of PruneFuse. We clarify the conflating intermediate synchronization with the final fusion step in the discussion above.
>
> **3.**	_Samples selected with active learning using small models transfer reasonably well to larger structurally similar models_ (agree)
>
> We agree this is an important message. Critically, PruneFuse enables this transfer in a scalable way by constructing the selector via structured pruning (providing a direct cost knob) and further proposes to reuse the selector’s training through weight-aligned fusion rather than discarding it.
>
> ---
> We believe the revisions improve the clarity and presentation of the work. We thank the reviewer again for the constructive and insightful feedback.
>
> ---
> ###### [1] Yuxia Sun, Aoxiang Sun, Siyi Pan, Zhixiao Fu, and Jingcai Guo. FedAPA: Server-side Gradient-Based Adaptive Personalized Aggregation for Federated Learning on Heterogeneous Data. In Proceedings of the 34th International Joint Conference on Artificial Intelligence (IJCAI '25), August 16–22, 2025, Montreal, Canada.
>
> ###### [2] Ozan Sener and Silvio Savarese. Active learning for convolutional neural networks: A core-set approach. The Sixth International Conference on Learning Representations, 2018.

---

> > ### Comment · Reviewer_Kwk8 · 2026-02-03
> >
> > I would like to thank the authors for providing numerous clarifications and improving the presentation.
> >
> > The rebuttal resolved most of my concerns:
> > 1. Description of the problem is significantly improved.
> > 2. The reformulated Algorithm 1 is easier to understand.
> >
> >    I would suggest changing AL to active learning to slightly improve readability.
> > 3. The caption of Figure 2 explicitly acknowledges that the presented trajectories serve as a conceptual illustration of faster convergence.
> > 4. I appreciate the detailed explanation on the "fusion vs without fusion" and on the meaning of synchronisation.
> >
> >    With this additional explanation I would change my claim from "fusion step does not lead to a significant increase in accuracy" to "intermittent prune + fuse steps lead to accuracy roughly similar to a single prune step (at the beginning) and a single fuse step (at the end)".
> >    I agree that Figure 4 indicates that the fuse step itself leads to an improved accuracy.
> > 5. The discussion on the importance of knowledge distillation is extended and now reflects the empirical finding much better.
> > 6. I agree that the authors provide a more thorough discussion of theoretical statements.
> >
> >    Still, I do not think presented theory is a core of the contribution or that it strongly reinforces empirical evidence. I do not see this as a major problem, because numerous experiments support the main claims of the authors well enough even with no theory at all.
> >
> > I would like to thank the authors for their thorough engagement with the feedback and for the clarity provided in their latest response. The revised manuscript represents a substantial improvement compared to the initial version and is now well-suited for publication in TMLR.

---

> > > ### Author Response · Authors · 2026-02-06
> > >
> > > We thank the reviewer for carefully reviewing the revisions and for the positive assessment. We appreciate the engagement and are glad that the clarifications and changes addressed the concerns raised.

---

### Review · Reviewer_V76W · 2026-01-07

**Summary Of Contributions:**

The authors propose PruneFuse, a framework for efficient data selection in Active Learning (AL). To address the computational scalability bottleneck of traditional AL , the method utilizes structurally pruned networks as lightweight data selectors. A key contribution is the network fusion mechanism, which transfers learned weights from the pruned selector back to the full model, providing an optimized initialization that accelerates convergence and improves generalization. The approach also integrates knowledge distillation and iterative synchronization. Extensive experiments across CIFAR, ImageNet, and NLP datasets demonstrate that PruneFuse significantly reduces computational costs  while outperforming state-of-the-art baselines like SVP and ALSE.

**Key Strengths:**

- Novelty: The "prune-then-fuse" paradigm effectively addresses the limitations of prior "proxy" methods (e.g., SVP), which discard the proxy model after selection, by reusing learned parameters to aid the target model.

- Efficiency: The method demonstrates substantial reductions in FLOPs and training time across various benchmarks while maintaining or exceeding baseline accuracy.

- Rigorous Evaluation: The submission includes comprehensive experiments across diverse architectures (ResNet, ViT, VDCNN) and tasks, supported by theoretical error bound analysis.

**Weaknesses:**

- Pipeline Complexity: The proposed workflow involves multiple stages which introduces higher implementation complexity compared to standard AL pipelines.

**Audience:**

Yes

**Audience Explanation:**

The paper addresses the critical trade-off between data efficiency and training costs in Deep Learning. The findings are highly relevant to researchers focusing on Efficient AI, Active Learning, and Model Compression. The proposed method offers a practical solution for training large models in resource-constrained environments, which is of broad interest to the TMLR community

**Broader Impact Concerns:**

No concerns.

**Claims And Evidence:**

Yes

**Claims Explanation:**

The claims are well-supported by empirical and theoretical evidence:

- Efficiency & Accuracy: Tables 1 and 2 clearly show that PruneFuse achieves comparable or superior accuracy to baselines with a fraction of the computational cost (e.g., 95% less computation on ImageNet-1K).


- Efficacy of Fusion: The ablation studies (Figs. 4, 5, 6) provide convincing visual evidence that the fusion mechanism significantly accelerates convergence compared to training without fusion.

- Theoretical Grounding: Theorem 5.1 provides a formal error bound, theoretically justifying the synchronization strategy.

**Requested Changes:**

I recommend the following adjustments to further strengthen the submission:

- Implementation Complexity Trade-off: While the computational savings are clear, the engineering overhead of the PruneFuse pipeline is non-trivial. I suggest adding a concise discussion in the main text weighing this implementation complexity against the performance gains.
- Justification for Unaltered Weights: Section 4.4 states that unaltered weights in the fused model serve as gateways to unexplored loss landscape regions. Please provide a more rigorous justification or specific ablation evidence (referencing Section 8.5 if applicable) comparing this strategy against alternative initializations (e.g., zero or random re-initialization) to empirically validate the claim that retaining initial weights specifically aids "exploration."
- Sensitivity to Pruning Criteria: The method relies on structured pruning. While Section 8.13 in the supplementary material covers this, a brief summary in the main text regarding the method's robustness to different pruning criteria would reinforce the claim of flexibility.

---

> ### Author Response · Authors · 2026-01-16
> **Author Response**
>
> We thank the reviewer for the thorough and constructive feedback, and for recognizing the novelty and empirical gains of PruneFuse. We have revised the manuscript to directly address the requested clarifications on pipeline overhead, initialization choices in fusion, and robustness to pruning criteria. All changes are highlighted in **blue** in the revised manuscript.
>
> >_1.	Implementation Complexity Trade-off: While the computational savings are clear, the engineering overhead of the PruneFuse pipeline is non-trivial. I suggest adding a concise discussion in the main text weighing this implementation complexity against the performance gains._
>
> **Response:** Thank you for the suggestion. We added a brief discussion in the main text (_Section 6.2: Main experiments_) clarifying the engineering overhead versus gains. In particular, while PruneFuse introduces additional components (structured pruning and fusion), their overhead is lightweight and incurred infrequently. As summarized in _Section 8.1_, pruning and fusion incur costs of $O(P log P)$ and $O(P)$, respectively are one-time (or infrequent) operations, both negligible compared to the repeated dense model training, which costs $O(\left|L\right|⋅N⋅T)$.
>
> In practice the proposed PruneFuse components are straightforward to implement, and their overhead is dominated by the savings from replacing the dense selectors with pruned ones. For example, on CIFAR-10 with ResNet-56 at $b=50$%, the selector computation is reduced by $75-96$% across pruning ratios while achieving higher accuracy (_Table 2_). Similar trends hold across datasets and architectures.
>
> >_2.	 Justification for Unaltered Weights: Section 4.4 states that unaltered weights in the fused model serve as gateways to unexplored loss landscape regions. Please provide a more rigorous justification or specific ablation evidence (referencing Section 8.5 (now 8.6) if applicable) comparing this strategy against alternative initializations (e.g., zero or random re-initialization) to empirically validate the claim that retaining initial weights specifically aids "exploration."_
>
> **Response:** Thank you for raising this point. To provide a more rigorous justification, we added an explicit ablation study in the supplementary (_Section 8.6, Table 18_) comparing different initialization strategies for the unaltered weights after fusion: retaining the original initialization, zero initialization, and random re-initialization. The results show that zero initialization degrades performance, whereas retaining the original initialization and random re-initialization achieve comparable and substantially better results. Importantly,  the observed performance gap between fusion and non-fusion training (_Section 8.6 Fig. 5-7_) suggests that the observed “exploration” benefit arises from preserving non-zero, stochastic degrees of freedom in the unfused portion of the model, and is not uniquely tied to retaining the specific original initialization itself. We updated _Section. 4.4_ accordingly and clarified that our default implementation uses random re-initialization for the unaltered weights.
>
> >_3.	Sensitivity to Pruning Criteria: The method relies on structured pruning. While Section 8.13 in the supplementary material covers this, a brief summary in the main text regarding the method's robustness to different pruning criteria would reinforce the claim of flexibility._
>
> **Response:** Thank you for the suggestion. We revised the manuscript to make this robustness claim explicit in the main text and the discussion (around _Table 7: Effect of Pruning Techniques and Pruning Criteria_) now summarizes this particularly.
>
> We believe these revisions strengthen the clarity and practicality of the paper, and we thank the reviewer again for the helpful suggestions.

---

### Decision · Action_Editor_QVYw · 2026-02-25

**Recommendation:** Accept as is

**Audience:**

Yes

**Audience Explanation:**

Yes. All reviewers agree the topic is relevant to TMLR’s audience. The work addresses a practical and timely problem: reducing computational cost in active learning, while maintaining performance. Researchers in active learning, efficient training, model compression, and scalable ML systems would likely find the findings useful.

**Claims And Evidence:**

Yes

**Claims Explanation:**

Yes. All three reviewers explicitly state that the main empirical claims are supported by convincing experimental evidence. They highlight:
1) Extensive experiments across multiple datasets and architectures.
2) Clear comparisons of accuracy and computational cost.
3) Controlled ablations isolating fusion and (optionally) distillation.
Some reviewers note that certain interpretive or theoretical claims (e.g., about loss landscape exploration or contraction assumptions) are stronger than the evidence fully supports, but the core performance and efficiency claims are convincingly validated.

Please make sure the final version incorporating all necessary changes mentioned in the rebuttal.